# Chlorine-initiated oxidation of n-alkanes under high NOx conditions: Insights into secondary organic aerosol composition and volatility using a FIGAERO-CIMS

Dongyu S. Wang and Lea Hildebrandt Ruiz

Department of Chemical Engineering, The University of Texas at Austin, Austin, TX 78756, USA

*Correspondence to*: Lea Hildebrandt Ruiz (lhr@che.utexas.edu)

**Abstract.** Chlorine-initiated oxidation of *n*-alkanes ($C_{8-12}$) under high nitrogen oxides conditions was investigated. Observed secondary organic aerosol yields (0.16 to 1.65) are higher than those for OH-initiated oxidation of $C_{8-12}$ alkanes (0.04 to 0.35). A High-Resolution Time-of-Flight Chemical Ionization Mass Spectrometer coupled to a Filter Inlet for Gases and AEROsols (FIGAERO-CIMS) was used to characterize the gas- and particle-phase molecular composition. Chlorinated organics were observed, which likely originated from chlorine addition to the double bond present on the heterogeneously produced dihydrofurans. A two-dimensional thermogram representation was developed to visualize composition and relative volatility of organic aerosol components using unit-mass resolution data. Evidence of oligomer formation and thermal decomposition was observed. Aerosol yield and oligomer formation were suppressed under humid conditions (35 to 67 % RH) relative to dry conditions (under 5 % RH). The temperature at peak desorption signal, $T_{max}$, a proxy for aerosol volatility, was shown to change with aerosol filter loading, which should be constrained when evaluating aerosol volatilities using the FIGAERO-CIMS. Results suggest that long-chain anthropogenic alkanes could contribute significantly to ambient aerosol loading over their atmospheric lifetime.

## 1 Introduction

Alkanes account for up to 90% of all anthropogenic hydrocarbon emissions and 12 % (140 Tg yr$^{-1}$) of annual non-methane hydrocarbon emissions (Fraser et al., 1997; Goldstein and Galbally, 2007; Guenther et al., 2012; Rogge et al., 1993; Schauer et al., 1999, 2002). Linear, branched, cyclic alkanes, and aromatics are major components of gasoline, diesel, motor oil, and other petroleum products (Caravaggio

et al., 2007; Kleeman et al., 2008; Schauer et al., 1999). Depending on its vapor pressure, the alkane can be emitted as volatile organic compound (VOC), intermediate-volatility organic compound (IVOC), semi-volatile organic compound (SVOC), or primary organic aerosol (POA). Evaporation of POA due to dilution can provide additional gas-phase alkanes, which can undergo photooxidation initiated by OH, NO$_3$, as well as chlorine radicals (Aschmann and Atkinson, 1995; Atkinson and Arey, 2003). Consequently, alkanes can have significant contributions to SOA production in urban environments (Dunmore et al., 2015). Hydrocarbon-like organic aerosol, which is often associated with POA and alkane oxidation, contributes on average 36 % to fine particulate matter (PM$_1$) in urban environments (Zhang et al., 2007). Influences of alkane emission and oxidation on SOA formation can be observed in remote regions as well (Carlton et al., 2010; Chrit et al., 2017; Hoyle et al., 2011; Minguillón et al., 2011, 2016; Patokoski et al., 2014; Saito et al., 2004; Sartelet et al., 2012). Alkane emissions can be sporadic and scattered, complicating both monitoring and modeling efforts (Lyon et al., 2015; Zavala-Araiza et al., 2015, 2017). Using laboratory results on OH-initiated SOA formation from alkanes under low- and high-NO$_x$ conditions (Jordan et al., 2008; Lamkaddam et al., 2017; Lim and Ziemann, 2009a, 2005, 2009b; Loza et al., 2014; Presto et al., 2009, 2010; Schilling Fahnestock et al., 2015; Takekawa et al., 2003; Tkacik et al., 2012; Yee et al., 2012, 2013; Zhang et al., 2014), model predictions for OA concentrations can be improved, though the predicated OA is often less oxidized than observed (Bahreini et al., 2012; DeCarlo et al., 2010; Dzepina et al., 2009; de Gouw et al., 2011; Murphy and Pandis, 2009; Shrivastava et al., 2008; Zhang et al., 2007), which could point to missing oxidants (e.g. Cl) and SOA oxidation mechanisms in the models (Murphy and Pandis, 2009).

Recent field studies have identified reactive chlorine compounds in diverse locales from natural and anthropogenic sources (Faxon and Allen, 2013; Finlayson-Pitts, 2010; Saiz-Lopez and von Glasow, 2012; Simpson et al., 2015). Tropospheric chlorine chemistry can enhance ozone production (Tanaka et al., 2003). Nighttime production of nitryl chloride mediated by heterogeneous uptake of NO$_x$ onto chloride-containing particles represents an ubiquitous source of reactive chlorine (Thornton et al., 2010). In addition, biomass burning could also act as a source of ClNO$_2$ (Ahern et al., 2018). ClNO$_2$ photolysis in the early morning produces Cl and NO$_x$, which has been shown to enhance RO$_2$ production from alkane oxidation in near coastal regions (Riedel et al., 2012) as well as OH radical propagation in urban

environments (Young et al., 2014). In addition to reactive chlorine emissions from water treatment (Chang et al., 2001) and fuel combustion (Osthoff et al., 2008; Parrish et al., 2009), the rising usage of volatile chemical products (VCP) such as pesticides, cleaning products, and personal care products may be a significant source of reactive chlorine compounds and VOCs in urban environments (Khare and Gentner, 2018; McDonald et al., 2018). VOC-Cl oxidation products such as isomers of 1-chloro-3-methyl-3-butene-2- (CMBO), a tracer for isoprene-Cl chemistry (Nordmeyer et al., 1997), have been observed in highly polluted environments (Le Breton et al., 2018; Tanaka et al., 2003).

Akin to OH radicals, Cl radicals initiate reactions with alkanes via hydrogen-abstraction, forming hydrogen chloride (HCl) and alkylperoxy radicals ($RO_2$). According to structure-activity relationships, terminal hydrogen abstraction occurs more frequently with alkane-Cl than with alkane-OH reactions (Kwok and Atkinson, 1995), resulting in different product distributions; for example, increased formation of primary alkyl nitrates. The biomolecular gas-phase reaction rate constants with linear alkanes for OH radicals and Cl radicals increase with the alkane chain length. Consequently, the reaction rate constants for Cl and linear $C_{8-12}$ alkanes range from $4.05 \times 10^{-10}$ (octane) to $5.36 \times 10^{-10}$ $cm^{-3}$ molecules$^{-1}$ s$^{-1}$ (dodecane) at 298 K and 1 atm (Aschmann and Atkinson, 1995), which is over an order of magnitude higher than the rate constants for OH radicals, $8.11 \times 10^{-12}$ (octane) to $1.32 \times 10^{-11}$ $cm^{-3}$ molecules$^{-1}$ s$^{-1}$ (dodecane; Atkinson and Arey, 2003). Studies show that Cl-initiated oxidation of volatile organic compounds such as isoprene, monoterpenes, toluene, and polycyclic aromatic hydrocarbons can lead to rapid SOA formation with high yields (Cai et al., 2008; Cai and Griffin, 2006; Huang et al., 2014; Karlsson et al., 2001; Ofner et al., 2013; Riva et al., 2015; Wang and Hildebrandt Ruiz, 2017). Compounds consistent with isoprene-derived organochloride were recently observed in filter samples collected in Beijing (Le Breton et al., 2018).

Although all initial $C_{8-12}$ alkane-Cl oxidation products are expected be non-chlorinated, formation of alkane-derived organochlorides is possible from Cl addition to multi-generational products. Studies show that the oxidation of alkanes with 5 or more linear carbons produces 1,4-hydroxycarbonyl that can undergo a rate-limiting, acid-catalyzed heterogeneous reaction to produce dihydrofuran (DHF) compounds (Atkinson et al., 2008; Holt et al., 2005; Jordan et al., 2008; Lim and Ziemann, 2009b, 2009c). DHF is highly reactive: For 2,5-dihydrofuran, the bimolecular reaction rate constants with $O_3$, OH and Cl

are $1.65 \pm 0.31 \times 10^{-17}$, $6.45 \pm 1.69 \times 10^{-11}$, and $4.48 \pm 0.59 \times 10^{-10}$ cm$^3$ molecule$^{-1}$ s$^{-1}$, respectively (Alwe et al., 2013, 2014). Chlorine radicals can react with dihydrofuran via both H-abstraction and Cl-addition, producing chlorinated (e.g. dichlorotetrahydrofurans) and non-chlorinated compounds (e.g. furanones) under low NO$_x$ conditions (Alwe et al., 2013). Similarly, formation of both chloronitrates (via Cl-addition) and organonitrates (via H-abstraction) from alkane-Cl oxidation is possible in the presence of NO$_x$. In this study, environmental chamber studies were conducted using long-chain (C$_{8-12}$) $n$-alkanes and Cl radicals under high NO$_x$ conditions to evaluate the Cl-initiated SOA formation from alkanes and to characterize the molecular composition of gas and aerosol compounds.

## 2 Methods

### 2.1 Environmental chamber experiments and instrumentation

Experiments were conducted inside a 10 m$^3$ Teflon® chamber at 298 K, using UV lights to generate radicals. The NO$_2$ photolysis rate was measured to characterize the UV intensity (Carter et al., 2005) and was found to be similar to ambient levels (0.53 min$^{-1}$ at 0° zenith angle) at approximately 0.5 min$^{-1}$. The chamber relative humidity (RH) ranged from 0 to 67 %. For dry experiments, the chamber was filled with dried clean air supplied by a clean air generator (model 737R, Aadco). Under typical atmospheric conditions, elevated RH can be expected, especially within the marine boundary layer in near-coastal regions, where Cl-alkane chemistry may be important (Riedel et al., 2012). Therefore, SOA formation under humid conditions was also investigated. For humid experiments (i.e. RH > 20 %), the chamber was flushed overnight with humidified clean air. To reduce wall-loss, dried ammonium sulfate seed particles were injected into the chamber. The seed particles were generated from a 0.01 M aqueous ammonium sulfate solution using an aerosol generation system (AGS 2002, Brechtel). Chlorine gas (101 ppm in N$_2$, Airgas) was injected as the Cl radical precursor. Each $n$-alkane ($n$-octane, 99 %; $n$-decane, 99 %; dodecane > 99%, Sigma-Aldrich) was first injected into a glass gas sampling tube (Kimble-Chase, 250 mL), which was flushed with gently heated clean air into the chamber at 2 lpm for at least 30 minutes. Precursor NO (9.98 ppm in N$_2$, Airgas) and NO$_2$ (9.86 ppm in N$_2$, Airgas) were injected into the chamber using a mass flow controller (GFC17, Aalborg). Initial concentrations of VOC and oxidant precursors are summarized

in Table 1. Gas-phase NO and $NO_2$ concentrations were monitored by a chemiluminescence monitor (200E, Teledyne). The $O_3$ concentration was monitored using a photometric ozone analyzer (400E, Teledyne). The start of photooxidation, initiated by turning on all the UV lights after the precursors were well-mixed in the chamber, was designated as the reference point (i.e. time 0 min) for each experiment. UV lights were turned off after 60 minutes. On the day before and after each SOA formation experiment, ammonium sulfate seed particles and 50 ppb of chlorine gas were injected into the chamber. UV lights were turned on to generate Cl radicals and remove residual reactive organic compounds in the chamber. Minimal SOA formation was observed during these cleaning experiments.

Particle size distributions were characterized using a scanning electrical mobility system (SEMS, Brechtel model 2002). The particle-phase bulk chemical composition was measured using an aerosol chemical speciation monitor (ACSM, Aerodyne). The ACSM was calibrated with 300nm size-selected ammonium nitrate and ammonium sulfate aerosols generated from nebulized 0.005 M solutions using the AGS to determine the nitrate response factor (RF) and relative ionization efficiencies (RIE) for ammonium and sulfate, which are required for ion-to-mass signal conversions. Using electron impact ionization, the ACSM can measure the submicron, non-refractory aerosol bulk composition at one minute intervals (Budisulistiorini et al., 2013; Ng et al., 2011a). Using a standard fragmentation table (Allan et al., 2004), the ACSM can speciate the aerosol content into organics, nitrate, sulfate, ammonium, and chloride (Ng et al., 2011a). The ability of the ACSM to detect organic chloride using $HCl^+$ (*m/z* 36) has been demonstrated previously for isoprene-Cl SOA (Wang and Hildebrandt Ruiz, 2017). The default $RIE_{Chl}$ value of 1.3 was used for chloride mass conversion. The $Cl^+$ (*m/z* 35) ion was excluded from chlorine quantification as it showed inconsistent response to non-refractory chlorides (e.g. ammonium chloride). ACSM data were analyzed in Igor Pro V6.37 (Wavemetrics, Inc.) using ACSM local v1603 (Aerodyne) and other custom routines. Organic aerosol concentrations were calculated using ACSM measurements assuming a collection efficiency of 0.5 and RIE of 1.4, corrected for depositional particle wall loss (Pathak et al., 2007) but not for organic vapor loss (Huang et al., 2018b; Krechmer et al., 2017; Nah et al., 2017).

## 2.2 FIGAERO-CIMS

A high-resolution time-of-flight chemical ionization mass spectrometer (CIMS, Aerodyne) was used to measure the gas-phase chemical composition using $I^-$ as the chemical ionization reagent. Humidified UHP $N_2$ was flushed over a methyl iodide permeation tube and then through a $^{210}Po$ ionizer into the ion-molecule reaction (IMR) chamber of the CIMS. Theory and operation of the CIMS are described in detail elsewhere (Aljawhary et al., 2013; Bertram et al., 2011; Lee et al., 2014; Wang and Hildebrandt Ruiz, 2017). The CIMS inlet was coupled to a filter inlet for gases and aerosols (FIGAERO), which has been used in a number of studies to investigate the chemical composition and volatility of particle-phase compounds (D'Ambro et al., 2017; Gaston et al., 2016; Huang et al., 2018a; Lee et al., 2016; Lopez-Hilfiker et al., 2014, 2015; Stark et al., 2017; Thompson et al., 2017). The FIGAERO system alternated between two operational modes. In the aerosol collection mode, a gas-aerosol mixture was drawn through a PTFE filter (Zefluor® 2.0 μm 24mm, Pall Corp.) at 3 SLPM for 15 to 45 minutes while gas species were sampled and analyzed. In the desorption mode, clean air or UHP $N_2$ was passed through the filter and heated to 200 ºC (measured just above the filter) at a rate of approximately 5 to 10 ºC min$^{-1}$ for 40 to 20 minutes, respectively. The filter was then soaked at 200 ºC for 20 minutes. The volatilized vapor was sampled by the CIMS, and the desorption signal as a function of temperature can be used to construct a one-dimensional (1-D) thermogram (e.g. Fig. 6a). For a monomeric compound, the desorption signal as a function of temperature is expected to be monomodal (Lopez-Hilfiker et al., 2014). The temperature at the peak desorption signal, $T_{max}$ correlates with the enthalpy of sublimation (dH$_{sub}$) and the saturation vapor pressure ($C^*$) of the compound (Lopez-Hilfiker et al., 2014, 2015). However, some compounds may exhibit bimodal or multimodal behavior, with a second desorption mode occurring at much higher temperatures than expected, which is often interpreted as the result of thermal decomposition of larger oligomeric compounds (Huang et al., 2018a; Lopez-Hilfiker et al., 2014; Stark et al., 2017; Wang et al., 2016).

CIMS and FIGAERO data analysis was conducted in Igor Pro with Tofware v2.5.10 (Tofwerk) and custom routines. Hundreds of ions belonging to diverse chemical families can be retrieved from the mass spectra. To simultaneously represent the chemical composition, relative aerosol volatility (i.e. $T_{max}$ distribution), and multimodal thermal desorption behaviors, a two-dimensional (2-D) thermogram

framework was developed. The 2-D thermogram is comprised of normalized unit-mass resolution 1-D thermograms, each expressed as a percentage color scale of the maximum desorption signal. 2-D thermogram applications are discussed in Section 3.2. The advantage of using UMR over HR data is the ability to investigate the SOA thermal desorption behavior over the entire $m/z$ and volatility (i.e. $T_{max}$) range without having to assign chemical formulae to all ion. This HR analysis is time-consuming especially for high molecular weight compounds whose exact molecular composition can be difficult to ascertain. The disadvantage of using UMR over HR data is the overlapping of ions and potential interference by isotopic signals or non-adduct ions. Therefore, the 2-D thermogram should be used as complement rather than a replacement of the HR analysis. Based on the SOA molecular composition as observed by the FIGAERO-CIMS, the average oxidation state of carbon ($OS_C$) may be estimated,

$$OS_C = 2 \times O{:}C - H{:}C + NO_3{:}C + Cl{:}C \qquad \text{Eq. (1)}$$

where $NO_3{:}C$, $Cl{:}C$, $O{:}C$, and $H{:}C$ are the molecular ratios of the number of $-NO_3$ functional groups, $-Cl$ functional groups, non-$NO_3$ oxygen atoms, and H atoms to the number of carbon atoms for any given compound. The average SOA $OS_C$ is calculated based on iodide-adducts only. For simplicity, all organic ions were assumed to have equal sensitivity, which is known to vary with ion cluster binding energy and sample RH (Hyttinen et al., 2018; Iyer et al., 2016; Lopez-Hilfiker et al., 2016).

## 3 Results and Discussion

### 3.1 SOA and organic chloride formation

Experimental conditions and results are summarized in Table 1. Chlorine-alkane SOA yields increased with VOC precursor length, consistent with the trend observed for OH-alkane SOA (Jordan et al., 2008; Lim and Ziemann, 2009b; Presto et al., 2010; Schilling Fahnestock et al., 2015; Yee et al., 2012, 2013). For similarly functionalized alkane oxidation products, the vapor pressure decreases roughly log-linearly as the precursor carbon number increases from 8 to 15 (Jordan et al., 2008; Presto et al., 2010). Comparison of the unit-mass ACSM spectra for octane (Exp. 3), decane (Exp. 7), and dodecane (Exp. 11) SOA in Fig. S1 shows a consistent increase in the fractional contributions to the bulk OA mass (i.e. $f_{m/z}$) by organic ions at $m/z$ 27 ($C_2H_3^+$), 41 ($C_3H_5^+$), 55 ($C_4H_7^+$) and 57 ($C_4H_9^+$) with increasing alkane length.

Select odd *m/z* ions, noticeably *m/z* 55 and 57, have been used as tracers for hydrocarbon-like organic aerosol (HOA), and sometimes for primary aerosol emissions as well (Ng et al., 2011b; Ulbrich et al., 2009). Given the same oxidation conditions, SOA products derived from longer alkane precursors appeared less oxidized, as seen by the decrease in $f_{44}$ (presumably mostly $CO_2^+$), but more hydrocarbon-like, as seen by the increase in $f_{57}$, consistent with previous observations (Lambe et al., 2012; Loza et al., 2014). The results are summarized in Table 1. As the alkane chain length increases, the increased SOA production also favors the partitioning of semi-volatile compounds into the particle-phase (Donahue et al., 2006; Pankow, 1994), further lowering $f_{44}$.

Time series of gas-phase NO, $NO_2$, $O_3$, $Cl_2$, and suspended particle-phase organics and chlorine concentrations during photooxidation are shown in Fig. 1 using dodecane-Cl oxidation under low RH (Exp. 11) as the representative example. The $Cl_2$ concentration was estimated using $I^-$ CIMS by tracking the $Cl_2I^-$ ion. In addition, ions consistent with $HO_2NO_2$, HONO, and $ClNO_2$ were observed in the gas phase, as shown in Fig. 1. The $ClNO_2I^-$ ions are expected to correspond to both $ClNO_2$ (< 20 %) and chlorine nitrite (ClONO, > 80 %) from the reaction between $Cl^\bullet$ and $NO_2$ (Golden, 2007; Niki et al., 1978). Formation of $HO_2NO_2$ from the reaction of $HO_2$ and $NO_2$ has been observed using $I^-$ CIMS previously (Veres et al., 2015). Formation of HONO was likely due to the interactions between $HO_x$ and NO (see R. S2 and R. S3 in the SI). Under UV, HONO decays to background level. Production of $HO_2NO_2$ and HONO is indicative of secondary $HO_x$ chemistry enabled by primary Cl-alkane oxidation chemistry, consistent with previous studies (Wang and Hildebrandt Ruiz, 2017; Young et al., 2014).

The NO concentration decreased sharply at the beginning of the photooxidation accompanied by increases in the concentrations of $NO_2$ and ozone. SOA production was most rapid during this initial period (0 to 10 min). Afterwards, at around 15 min in Fig. 1, the concentrations of $NO_2$ and ozone stabilized. Oxidation continued under UV driven by Cl radicals, and the SOA concentration began to decay due to oxidative fragmentation (Kroll et al., 2011; Lambe et al., 2012; Wang and Hildebrandt Ruiz, 2017) and wall loss. Ozone production continued slowly under UV at ~0.24 ppb min$^{-1}$. Slight $NO_2$ production (< 0.1 ppb min$^{-1}$) was also observed, which may be due to production of nitrous acid (HONO) on the Teflon® surface, a common background contaminant in environmental chambers (Carter et al., 2005; Rohrer et al., 2004).

SOA production was dependent on the initial NO and $NO_2$ concentrations, as shown in Fig. S2. Higher initial NO concentrations led to higher SOA yields and lower ozone production for all precursors, as shown in Table 1 and Fig. S2. This is similar to alkane-OH SOA formation, where higher NO concentrations lead to more abundant organic nitrate formation, which increases the SOA volume (Schilling Fahnestock et al., 2015) and density (Loza et al., 2014). This is consistent with FIGAERO-CIMS results where the particulate organonitrate molar ratio (calculated as the average number of $-NO_3$ per molecule) and mass fraction were the highest in SOA produced under NO-only conditions (see Table S1). The organonitrate molar ratio also increased with the alkane precursor length, from 0.57-0.64 for octane to 0.75-0.93 for dodecane. A similar trend was observed for the orgnaonitrate mass fraction, which increased from 0.53-0.58 for octane to 0.66-0.72 for dodecane. The alkane-Cl SOA yields, ranging from 0.16 (octane) to 1.65 (dodecane), are 4 times higher than the alkane-OH SOA yields (0.04 for octane and 0.35 for dodecane) that were obtained using 1 ppm alkane, 10 ppm methyl nitrite, and 10 ppm NO (Lim and Ziemann, 2009a), much higher than the precursor concentrations used here (13 to 15 ppb alkanes, < 40 ppb $NO_x$, 40 ppb $Cl_2$). Formation of primary alkyl nitrates due to terminal H-abstraction by Cl could contribute to the higher SOA yields observed for alkane-Cl than for alkane-OH oxidation, as primary alkyl nitrates have lower saturation vapor pressures than secondary alkyl nitrates (Lim and Ziemann, 2009b; Yeh and Ziemann, 2014, 2015). The apparent SOA density, calculated using the ACSM SOA mass measurement and the SEMS SOA volume measurement, was $2.1 \pm 0.3$ g cm$^{-3}$, which is substantially higher than that reported for OH-alkane SOA formed under high $NO_x$ conditions, 1.06 to 1.28 g cm$^{-3}$ (Lim and Ziemann, 2009a; Loza et al., 2014). These differences could be due to uncertainties associated with the ACSM and similar instruments, specifically the RIE (Li et al., 2018; Xu et al., 2018) and CE (Docherty et al., 2013; Middlebrook et al., 2012; Robinson et al., 2017), which may vary with SOA composition, oxidation state, or phase state. Assuming an SOA density of 1.06 g cm$^{-3}$, the SOA yield (0.10 to 0.99) is still higher for Cl-alkane oxidation as compared to OH-alkane oxidation. Molar SOA yields, calculated using the average molecular weight of species identified with the FIGAERO-CIMS, are summarized in Table S1.

Although the initial Cl-alkane reactions proceed via a H-abstraction pathway, particulate chlorine was observed using the ACSM as shown in Fig. 1. Direct halogenation of the $C_{8-12}$ alkyl radicals is expected

to be minimal, given the low amounts of $Cl_2$ present (maximum of 40 ppb). A carbon-carbon double bond is required to enable Cl-addition reactions and organic chlorine formation. The heterogeneous production of dihydrofuran (DHF) via 1,4-hydroxycarbonyl uptake, acid-catalyzed isomerization, and dehydration reactions (Atkinson et al., 2008; Jordan et al., 2008; Lim and Ziemann, 2009c, 2009b), followed by Cl addition to the DHF double bond, could be the source of observed organic chlorine. A condensed reaction pathway is shown in Fig. 2. The delay in particulate chlorine formation relative to that of bulk organics, as shown in Fig. 1, is consistent with the rate-limiting heterogeneous DHF production (Holt et al., 2005; Zhang et al., 2014). Under high $NO_x$ conditions, the peroxy radical product from the chlorine-addition pathway could react with NO to form a cyclic hemiacetal chloronitrate or an alkoxy radical that would undergo ring-opening reactions. Compounds resembling chloronitrates (e.g. $ONO_2$-$C_{12}H_{18}ClO_3 \cdot I^-$ for dodecane) were tentatively identified in the particle-phase using the FIGAERO but they were not well separated from the shoulder of nearby organonitrate peaks (e.g. $ONO_2$-$C_{12}H_{21}O_5 \cdot I^-$), as shown in Fig. S9. Select $C_{2\text{-}6}$ organochlorides were well-separated from nearby peaks and confirmed by the distinct Cl isotopic signal, the thermograms of which are shown in Fig. S10. The most abundant gas-phase organic chlorine compounds observed were products of ring-opening reaction pathways such as $C_2H_3ClO_2$ and $C_{4\text{-}6}$ compounds. Small amounts of $C_2H_2Cl_2O_2$ were also observed.

Organochloride formation is expected to be lower under humid conditions, where DHF formation is inhibited (Holt et al., 2005; Zhang et al., 2014; Ziemann, 2011). Evidence consistent with organochloride suppression under humid condition was observed for dodecane SOA only, where the organochloride (including chloronitrates) mass fraction decreased from 0.15 (Exp. 11, < 5 % RH) to 0.13 (Exp. 12, 67 % RH) as measured by the FIGAERO-CIMS. The mass fraction of the -Cl functional group decreased from $1.8\ E^{-2}$ to $1.6\ E^{-2}$ as measured by the FIGAERO-CIMS (Table S1) or from $1.4\ E^{-2}$ to $1.1\ E^{-2}$ as measured by the ACSM (Table 1) as the RH increased. No clear differences were observed for octane or decane SOA, which may be due to the less extreme RH conditions investigated, uncertainties with organochloride ion identification in the CIMS, or the lower organochloride concentrations observed in Exps.1-8, which is especially challenging for chloride quantification using the ACSM.

DHF ozonolysis can compete with chlorination. For instance, under typical marine boundary layer (MBL) conditions, the chlorine-initiated oxidation was estimated to be a significant sink of 2,5-DHF,

accounting for 29 % of the reaction in the presence of OH and $O_3$ (Alwe et al., 2014). In contrast, only 1.8 % of 2,3-DHF consumption was attributed to Cl radical chemistry in the MBL, owing to the increased reactivity of 2,3-DHF (relative to 2,5-DHF) towards ozone and OH radicals (Alwe et al., 2014). The reported alkane-OH reaction mechanisms expect the formation of substituted 2,3-DHF (Ziemann, 2011).

The alkane-derived organochloride yield under ambient conditions is therefore expected to be smaller in the presence of elevated RH and $O_3$ levels. Although ozone can impose an upper limit on alkane SOA yields and oxidation state, suppressing the multigenerational OH-alkane oxidation chemistry (Zhang et al., 2014), the continued gas-phase processing of OH-alkane and DHF-ozonolysis products via H-abstraction by chlorine radicals could counteract these limitations.

The gas-phase alkane-Cl oxidation products formed under high $NO_x$ conditions were dominated by organonitrates, which was also reflected in the dominance of organonitrates in the particles phase (see Table 1 and Section 3.2). Figure 3 shows the formation of early generation (e.g. hydroxynitrates, $ONO_2\text{-}C_nH_{2n+1}O$) and multi-generational (e.g. oxidized organonitrates, $ONO_2\text{-}C_nH_{2n-5}O_4$) oxidation products from different alkane precursors. Like the bulk particle phase composition, gas-phase

compounds derived from smaller alkane precursor were more oxidized given similar oxidation conditions (e.g. NO, $NO_2$, and $Cl_2$): the signal of oxidation products with similar oxygen numbers (and higher oxidation state) peaked earlier into the photooxidation period for smaller precursors (e.g. $ONO_2\text{-}C_8H_{11}O_4$ vs. $ONO_2\text{-}C_{12}H_{19}O_4$ in Fig. 3). As oxidation continued, driven primarily by gas-phase chemistry (Aimanant and Ziemann, 2013), the importance of fragmentation reactions increased relative to that of

functionalization reactions (Lambe et al., 2012). The heterogeneous oxidation of SOA (Bertram et al., 2001; George and Abbatt, 2010), which is expected to drive the oxidation of very large (n>30) alkanes (Lim and Ziemann, 2009b), may also contribute to oxidation and fragmentation observed here, but its impacts are beyond the scope of this work. Assuming a uniform CIMS sensitivity, Figure 4a shows that the gas-phase was dominated by $C_3$ to $C_5$ organic nitrates. Figure 4b shows that the $OS_C$ increased while

$n_C$ decreased as photooxidation continued, indicative of fragmentation reactions. In addition to hydroxynitrates and hydroxycarbonyl nitrates, dinitrates and trinitrates were also observed. The CIMS was not sensitive towards simple, alkane-derived alkyl nitrates ($ONO_2\text{-}C_nH_{2n+1}$). The lack of sensitivity

of the I$^-$ reagent ion towards alkyl and keto nitrates has been reported previously for isoprene and monoterpene-derived organic nitrates (Lee et al., 2016).

## 3.2 Two-dimensional thermogram

The FIGAERO filter spectra are shown in Fig. 4 for octane (Exp 3), decane (Exp. 7), and dodecane (Exp. 11). A log-scale version of Fig. 4 is shown in Fig. S3. The spectra are calculated from the average desorption ion signals observed when the filter temperature was between 40 and 140 $^o$C, As shown below and in Fig. 5, SOA components desorbed most effectively in this temperature range, with most organic ions having T$_{max}$ within this temperature range. At desorption temperature above 140 $^o$C, inorganic ions began to dominate the spectra. The C$_{8-12}$ alkane-Cl SOA share many similarities towards the lower $m/z$ (< 320) range which consisted of C$_{\leq 7}$ oxidized organic compounds. For larger alkane precursors, the particle phase composition was dominated by C$_n$ organic nitrates, which are grouped by their degree of oxygenation (i.e. number of non-nitrate O atoms), "O$_x$", as shown in Fig. 4c. O$_1$ to O$_5$ mono-nitrates, O$_2$ to O$_4$ dinitrates, and O$_1$ to O$_3$ trinitrates dominated each oxygenation group. O$_6$ to O$_8$ mononitrates were present in the particle-phase but were less abundant than the nearby O$_2$ to O$_4$ dinitrates (e.g. (ONO$_2$)$_2$-C$_{12}$H$_{20}$O$_2$ at $m/z$ 447 > (ONO$_2$)$_2$-C$_{12}$H$_{22}$O$_2$ at $m/z$ 449 > (ONO$_2$)-C$_{12}$H$_{17}$O$_6$ $m/z$ 446 > (ONO$_2$)-C$_{12}$H$_{19}$O$_6$ at $m/z$ 448). An example of a C$_{12}$ mono-nitrate distribution is shown in Fig. S4. Assuming the same sensitivity towards the different organic nitrates observed, the organic nitrate abundance follows a bell-shaped distribution, similar to field observations for C$_5$ and C$_{10}$ organic nitrates derived from isoprene and monoterpenes, respectively (Lee et al., 2016). For dodecane mononitrates, the abundance peaked around O$_3$ and O$_4$ and decreased towards O$_2$ and O$_5$, as shown in Fig. S4. Dinitrate abundance decreased from O$_2$ to O$_4$. Trinitrate decreased from O$_1$ to O$_3$. Similar trends can be observed for octane and decane SOA. As the precursor length increased, the SOA appeared less oxidized, as shown in Fig. 4d, consistent with ACSM observations for $f_{44}$ (see Table 1 and Fig. S1), which is correlated with OS$_C$ (Canagaratna et al., 2015). Simple hydroxynitrates (e.g. ONO$_2$-C$_n$H$_{2n+1}$O) were not observed in the particle phase as they were completely oxidized in the gas-phase, as shown in Fig. 3.

In addition to the identification of aerosol chemical composition, the FIGAERO-CIMS can be used to estimate the aerosol volatility using empirical correlations between C$^*$ and T$_{max}$ (Lopez-Hilfiker et al.,

2014, 2015). The particle-phase chemical composition and volatility distribution can be represented in several ways. For instance, desorption signals of select compounds can be normalized against the maximum and plotted on the same 1-D thermogram (ion signal versus filter temperature) to compare their relative volatilities (Lopez-Hilfiker et al., 2014). Scatter plots of oxygen number or O:C ratio versus carbon number colored by $T_{max}$ or particle-phase mass fraction have been used to show changes in aerosol composition (Huang et al., 2018a; Lopez-Hilfiker et al., 2014, 2015). To show the differences of various types of thermal desorption products (i.e. monomers, thermal decomposition products, and oligomers), averages of the 1-D thermograms can be shown (Huang et al., 2018a; Lopez-Hilfiker et al., 2015). Scatter plots of $T_{max}$ or the $OS_c$ versus molecular weight have been used to distinguish monomers and oligomers for select compounds (Wang et al., 2016). Here we propose a new framework to represent aerosol composition, relative volatility, and oligomer decomposition simultaneously using 2-D thermograms shown in Fig. 5a-e.

As illustrated in Fig. 5a for octane-chlorine SOA, the thermal desorption products can be separated into 5 different groups. Region 1 ($m/z < 350$, $40 < T_{max} < 90\ ^oC$) was composed of a group of semi-volatile compounds with similar $T_{max}$ values. This region also includes iodide-adducts which correspond to species that are too volatile to be present as molecular compounds in the particle phase and are likely low-temperature decomposition products. Prominence of ions smaller than $m/z$ 127 including $Cl^-$ and a range of organic ions, could be acid exchange or charge transfer products (as opposed to $I^-$ adducts) of low-temperature decomposition products. Although $I^-$ is a relatively soft ionization method compared to electron impact ionization, collision-induced ion fragmentation and cluster dissociation cannot be ruled out as a potential source for non-adducts. Region 2 ($350 < m/z < \sim700$, $40\ ^oC < T_{max} < 120\ ^oC$) consisted of monomers, dimers, and some oligomers. Oligomerization may proceed via condensed-phase reactions between cyclic hemiacetal compounds, forming acetal dimers, or reactions between cyclic hemiacetal and 1,4-hydroxycaronbyl compounds, forming hemiacetal dimers (Aimanant and Ziemann, 2013). Condensed-phase, acid-catalyzed isomerization of hemiacetal and acetal dimers is also possible (Aimanant and Ziemann, 2013). For OH-initiated oxidation of dodecane under high $NO_x$ conditions, hemiacetals and nitrate hemiacetals are reported to dominate the oligomer composition (Schilling Fahnestock et al., 2015), where nitrate hemiacetals could form via the reactive uptake of hydroxy

dihydrofuran and hydroxycarbonyl nitrate compounds (Schilling Fahnestock et al., 2015). The three oligomerization pathways are illustrated in Fig. S11. Ions consistent with acetal, hemiacetal, and nitrate hemiacetal compounds were observed in the particle phase using the FIGAERO-CIMS in regions 2 and 3. Oligomer formation via aldol condensation by highly oxidized compounds is also possible (Schilling

Fahnestock et al., 2015). Oligomerization may also proceed through a $NO_3$-elimination, reverse esterification process involving two alkyl nitrate compounds to form a singly nitrate oligomer, which was recently proposed for α-pinene SOA, though the exact mechanism remains elusive (Faxon et al., 2018).

An overall positive correlation between $T_{max}$ and molecular weight was observed in region 2. A similar correlation was also reported for SOA derived from oleic acid (Wang et al., 2016) and from α-pinene

(Faxon et al., 2018). In other words, overall, volatility decreased with molecular weight for compounds in region 2. However, within small $m/z$ segments in region 2, the correlation was not strictly linear and the $T_{max}$ varied cyclically. When the carbon and oxygen numbers were fixed (i.e. fixed O:C), the $T_{max}$ decreased (i.e. volatility increased) with increasing H:C (i.e. decreasing $OS_c$), as shown in Fig. 5e for $C_{12}$ organic nitrates (e.g. $ONO_2$-$C_{12}H_{19-25}O_3 \cdot I^-$ from $m/z$ 400 to 406) observed in dodecane-Cl SOA

desorption. Although decreases in $OS_c$ are often associated with increases in the aerosol volatility (Donahue et al., 2011; Kroll et al., 2011), fragmentation notwithstanding, the increase in H:C in each fixed O:C ion group was expected to decrease aerosol volatility. The increase in H:C was likely due to increased hydroxy functionalization over ketone functionalization, which should result in lowering of the saturation vapor pressure according to group contribution theory (Pankow and Asher, 2008). In addition,

the $T_{max}$ decreased from $O_3$:$C_{12}$ organonitrates (e.g. $ONO_2$-$C_{12}H_{21}O_3$, $T_{max}$ 95 $^oC$) to $O_4$:$C_{12}$ organonitrates (e.g. $ONO_2$-$C_{12}H_{21}O_4$, $T_{max}$ 89 $^oC$). The inconsistencies observed within small $m/z$ segments may be due to loading-dependent $T_{max}$ shift (see Section 3.3), where $T_{max}$ increases with mass loading. As shown in Fig. 4, the organonitrate ion intensities decreased from $O_3$ onwards. Overlapping of monomer and oligomer desorption peaks could also increase the apparent $T_{max}$.

Region 3 ($T_{max} > 90$ $^oC$) consisted of thermal decomposition products across a wide $m/z$ range. Select ions were present in both region 2 and region 3, resulting in bimodal thermal desorption, which can complicate $T_{max}$ estimation using the 2-D thermogram, especially when ions from the two desorption modes overlap. The thermal decomposition products may be the result of reversible dissociation of

oligomers (Faxon et al., 2018; Schobesberger et al., 2018) or the cleavage of C-O or C-C bonds, which has been observed in the thermal decomposition of monomeric organic acids (Stark et al., 2017). Additionally, SOA may exist in a "glassy" state (Koop et al., 2011), trapping volatile components (Perraud et al., 2012) that are released as the SOA mass is depleted. Region 4 (150 $^o$C < $T_{max}$ < 170 $^o$C) consists of mostly thermal decomposition products of the $(NH_4)_2SO_4$ seed particles. There was a clear gap between region 3 and 4. It is possible that the edge of regions 2 and 3 towards high $T_{max}$ range is the FIGAERO-CIMS's volatility detection limit for alkane-Cl SOA, where the energy required for vaporization exceeds that for thermal decomposition (Lopez-Hilfiker et al., 2014). Plateauing of $T_{max}$ towards the high $m/z$ range of region 2 appeared to agree with this hypothesis, where further increases in molecular weight, carbon number, and the degree of oxidation translated to negligible $T_{max}$ increase. Region 5 ($m/z$ > 700) likely consisted of high molecular weight, low volatility oligomers. Region 5 therefore likely consisted of $C_{24-36+}$ (dimers and above) compounds. Thermal desorption of some high molecular weight compounds at temperature much lower than expected ($T_{max}$ ~ 80 $^o$C) was observed for decane SOA and dodecane SOA. No definitive molecular composition was determined for these ions due to low signal intensity and wide range of potential chemical formulae.

The region boundaries can vary depending on the SOA composition as well as mass loading, as will be discussed in Section 3.3. The distribution and general features of the regions are consistent for alkane-Cl SOA formed in different experiments (Fig. 5 and Fig. S8). Comparison of 2-D thermograms for octane-Cl SOA (Fig. 5a) and dodecane-Cl SOA (Fig. 5b) shows that the dodecane SOA contains unique, high molecular weight oligomers ($m/z$ > 750), which are accompanied by significantly stronger thermal decomposition features in region 3. Bimodal thermal desorption behavior was observed, as shown in Fig. 5d, which masked the volatility-$m/z$ dependence expected for region 2. Compared to SOA formed under dry conditions (Fig. 5b, Exp 11), SOA formed under humid conditions (Exp. 12, 67 % RH) contained less high molecular weight oligomers in region 5 and exhibited much less thermal decomposition behavior in region 3, as shown in Fig. 5e, where a more distinct $T_{max}$-$m/z$ correlation could be established as compared to Fig. 5d. RH-induced oligomer suppression has been reported for toluene and α-pinene SOA formation (Hinks et al., 2018; Huang et al., 2018a). The oligomer decrease under high RH could also be due to RH-volatility dependent organic vapor wall loss (Huang et al., 2018b). A clear

*m/z* dependence of signal reduction under humid conditions as compared to the dry conditions is shown in Fig. S5. Additionally, SOA yield decreased (37 % for decane and 22 % for dodecane) under humid conditions while ozone production increased, which could be related to humidity-induced inhibition of 1,4-hydroxycarbonyl uptake and heterogeneous DHF formation (Holt et al., 2005). DHF is a key intermediate product for multigenerational oxidation chemistry and is reactive towards ozone (Zhang et al., 2014; Ziemann, 2011). Reduction in SOA yields under humid conditions has also been reported for dodecane-OH SOA formation (Schilling Fahnestock et al., 2015). The combined effects of DHF reduction and ozone enhancement would suppress organic chloride formation, which was observed for dodecane SOA, as discussed in Section 3.1.

### 3.3 $T_{max}$ shift

As the desorption temperature increased above 140 $^o$C, all dominant ions observed shared the same $T_{max}$, except for some background ions (e.g. $IHNO_3^-$), as shown in Fig. 5a-c, which were identified to be related to the thermal decomposition of $(NH_4)_2SO_4$ seed particles. From this process, $H_2SO_4$ vapor molecules produced were either detected as $H_2SO_4 \cdot I^-$ or were deprotonated to produce $HSO_4^-$, acting as a secondary chemical ionization reagent. Examples of 1[st] order (e.g. $HSO_4^-$), 2[nd] order (e.g. $H_2SO_4 \cdot HSO_4^-$) and 3[rd] order sulfate (e.g. $(H_2SO_4)_2 \cdot HSO_4^-$) clusters are shown in Fig. S6. It is worth noting that while the $T_{max}$ of sulfate ions was uniform within each filter desorption run, the $T_{max}$ varied between filter runs, which can be seen in Fig. 5a-c to be between 146 $^o$C and 153 $^o$C. Characterization experiments suggested that $T_{max}$ may increase with filter loading, as shown in Fig. 6a for pure levoglucosan aerosols and in Fig. S7a for pure $(NH_4)_2SO_4$ aerosols Whereas the $T_{max}$-mass loading correlation appeared to be linear for pure $(NH_4)_2SO_4$, as shown in Fig S7b. for $T_{max}$ between 136-149 $^o$C, a roughly sigmoidal correlation was observed for pure levoglucosan, as shown in Fig. 6b, where $T_{max}$ increased quickly from 43 to 62 $^o$C as filter loading increased from 0.2 to 0.8 µg.

$T_{max}$ shift phenomena have been observed in several FIGAERO-CIMS studies (D'Ambro et al., 2017; Gaston et al., 2016; Huang et al., 2018a; Lopez-Hilfiker et al., 2015; Thompson et al., 2017). Explanations vary for the observed $T_{max}$ shifts. For example, pinonic acid ($C_{10}H_{16}O_3$) identified in α-pinene ozonolysis SOA had a much higher $T_{max}$ (40 $^o$C) than expected from calibration ($T_{max} < 32$ $^o$C), which was attributed

to interactions between pinonic acid and other SOA components, causing a decrease in the measured apparent vapor pressure (Lopez-Hilfiker et al., 2015). However, no matrix effects were reported for the pinonic acid $T_{max}$ value (now around 65 °C instead of < 32 °C) obtained from a synthetic mixture of organic acid calibrants (Thompson et al., 2017), where some interactions between different organic molecules might have been expected. Whereas the pure compounds are shown to have sharp monomodal desorption peaks (Lopez-Hilfiker et al., 2014), the field campaign-average thermograms for individual compounds have broader desorption peaks (Thompson et al., 2017), which could be due to the presence of isomers (Thompson et al., 2017), variations in filter loading, differences in aerosol viscosity (Huang et al., 2018a), or interferences from oligomer decomposition products (Lopez-Hilfiker et al., 2014). Another $T_{max}$ shift example was reported for ambient biomass burning measurements, where the levoglucosan $T_{max}$ varied between 58 to 70 °C, in range of the 61.5 °C literature value (Lopez-Hilfiker et al., 2014) and the values observed here (41-64 °C),  between 5 pm to 10 am over the course of the campaign (Gaston et al., 2016). Between 10 am to 7 pm, the levoglucosan $T_{max}$ was significantly higher at 100 °C, which was attributed to thermal decomposition of oligomers produced from acid-catalyzed heterogenous reactions, as indicated by the increase in sulfate ion intensities during the same period (Gaston et al., 2016). It was not clear if the sulfate ions measured were derived from sulfuric acid or from the decomposition of $(NH_4)_2SO_4$, which could change the assumption made for aerosol acidity. A $T_{max}$ shift was also observed for isoprene SOA, where the mass and $T_{max}$ of non-nitrated OA decreased with increasing $NO_x$ concentration  (D'Ambro et al., 2017).  A recent investigation reported increases in $T_{max}$ for α-pinene ozonolysis SOA produced under dry conditions compared to SOA produced under humid conditions (Huang et al., 2018a). A positive correlation between aerosol loading and $T_{max}$ was reported for the first time, and the uniform thermal desorption peak shape assumption was questioned (Huang et al., 2018a). The loading dependence was reported to plateau at around 2 to 4 µg (converted from CIMS ion intensity assuming maximum sensitivity, i.e. that of formic acid), similar to observations for levoglucosan standards as shown in Fig. 6b, possibly due to saturation effects. Because increased oligomer content was observed under dry conditions, the authors suggested that viscosity effects were responsible for the observed $T_{max}$ shift (Huang et al., 2018a). It should be noted that plateauing of the $T_{max}$-loading dependence was demonstrated using averaged values, where ions were lumped based on carbon number.

Those that had less than or equal to 10 carbons were designated as monomers, which likely included desorption ions across multiple desorption regions (see Fig. 5). This categorization was perhaps too broad, and the loading dependence of individual ions could be lost after averaging. In contrast, the $T_{max}$ of oligomers, which were defined as compounds that contained more than 10 carbons—thus a more precisely defined group—only began to plateau at higher concentrations.

The $T_{max}$ mass dependence was also observed for alkane-Cl SOA. For Exp.6, three filters were collected and analyzed. The first filter was collected at 3 SLPM for 45 mins, followed by the second filter at 3 SLPM for 30 mins, and afterwards the third filter at 3 SLPM for 15 mins. The ratios of the unit-mass integrated ion signals during the desorption period for the first two filters relative to that of the third filter are shown in Fig. S8a. The associated 2-D thermograms are shown in Fig. S8b-d. Compounds in the $T_{max}$-$m/z$ dependence region (Region 2) showed a 10 to 20 °C $T_{max}$ increase from the lowest filter loading to the highest filter loading conditions. $T_{max}$ of sulfate-related ions increased from 149 °C to 172 °C with filter loading. The increase of enhancement ratio with $m/z$, as shown in Fig. S8a, reflects the changes in the composition of suspended aerosol in the chamber over time. Between the first (high-loading) filter collection and the third (low-loading) filter collection, volatility-dependent vapor wall loss may lead to a disproportionate decrease in high molecular weight, low volatility compounds (Huang et al., 2018b; Krechmer et al., 2016), resulting in increasingly greater enhancement towards the high $m/z$ region shown in Fig. S8a. The correlation between $T_{max}$ shift and integrated desorption signal for organics was not linear, unlike for ammonium sulfate, which may be due to matrix or saturation effects as previously suggested (Huang et al., 2018a). $T_{max}$-loading dependence may differ for individual FIGAEROs due to design variations or artifacts, such as sample distribution within the filter matrix or non-uniform heating, which would result in different aerosol evaporation kinetics (Schobesberger et al., 2018). Overall, our results show that the $T_{max}$ for organic and inorganic compounds varies with loading, which needs to be accounted for when estimating aerosol component volatility from the empirical correlations between $T_{max}$ and $C^*$ using the FIGAERO-CIMS.

# 4 Conclusion

Environmental chamber experiments were carried out to investigate the chlorine-initiated oxidation of $C_{8-12}$ linear alkanes under high $NO_x$ conditions. Rapid SOA formation and ozone production were observed. SOA yields increase with precursor length, consistent with alkane-OH SOA formation. Under similar oxidation conditions, alkane-Cl SOA exhibited more hydrocarbon-like characteristics as the alkane precursor length increased, as indicated by the ACSM measurements. This bulk SOA observation is consistent with gas- and particle-phase CIMS measurements, which identified more oxidized reaction products derived from smaller alkane precursors. CIMS measurements also suggest that organonitrates dominated the gas- and particle-phase composition. Trace amounts of alkane-derived organochlorides were observed using the ACSM and the FIGAERO-CIMS, likely produced via chlorine-addition to the heterogeneously produced dihydrofuran compounds. Organochloride and bulk SOA production were suppressed under humid conditions, whereas ozone production increased. Under such conditions, the SOA yields observed for chlorine-initiated oxidation of octane (0.24), decane (0.50), and for dodecane (1.10) were still much higher than those observed for OH-initiated oxidation of the respective alkanes. Overall, these results show that chlorine-alkane oxidation could be an important pathway for SOA production and ageing, especially in highly polluted environments replete with alkane, $NO_x$, and reactive chlorine emission sources.

A clear mass loading dependence for $T_{max}$ from FIGAERO-CIMS data was demonstrated using levoglucosan, ammonium sulfate, and alkane-Cl SOA, indicating that the quantitative assessment of $C^*$ from $T_{max}$ using the FIGAERO-CIMS needs to account for variations in filter loading. A unit-mass-resolution 2-D FIGAERO-CIMS thermogram framework was developed. The 2D thermograms demonstrate a clear relationship between molecular weight and volatility of aerosol components, and RH-induced suppression of oligomer formation. When used in conjunction with high-resolution ion fitting, the 2D thermogram can be a powerful tool for interpreting the chemical composition and volatility distribution of particle-phase compounds.

## Acknowledgements

This material is based upon work supported by the Welch Foundation under Grant No. F-1925 and the National Science Foundation under Grant No. 1653625. We thank P.J. Ziemann and L.B. Algrim for helpful discussions on the formation of organochlorides. We thank the Air & Waste Management Association for the Air Quality Research and Study Scholarship.

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

# Table and Figures

**Table 1. Summary of Experimental Conditions and Results**

| Exp # | VOC | NO[a] | NO$_2$[a] | Cl$_2$[a] | RH | SOA | Y$_{SOA}$[b] | $f_{44}$[c] | $f_{57}$[d] | f$_{HCl+}$[e] | O$_3$[f] |
|---|---|---|---|---|---|---|---|---|---|---|---|
| 1 | Octane | 35 | 2 | 40 | 5 > | 19.5 | 0.28 | 1.1 E$^{-1}$ | 1.4 E$^{-2}$ | 1.1 to 1.3 E$^{-2}$ | 56 |
| 2 | Octane | 1 | 36 | 40 | 5 > | 11.0 | 0.16 | 1.0 E$^{-1}$ | 1.6 E$^{-2}$ | 0.7 to 1.1 E$^{-2}$ | 69 |
| 3 | Octane | 17 | 19 | 40 | 5 > | 16.6 | 0.24 | 1.5 E$^{-1}$ | 1.1 E$^{-2}$ | 0.8 to 1.0 E$^{-2}$ | 56 |
| 4 | Octane | 17 | 19 | 40 | 35 | 16.8 | 0.24 | 1.0 E$^{-1}$ | 1.3 E$^{-2}$ | 1.1 to 1.4 E$^{-2}$ | 60 |
| 5 | Decane | 32 | 0 | 40 | 5 > | 68.3 | 0.84 | 1.0 E$^{-1}$ | 1.6 E$^{-2}$ | 0.7 to 1.1 E$^{-2}$ | 53 |
| 6 | Decane | 0 | 34 | 40 | 5 > | 43.1 | 0.45 | 7.3 E$^{-2}$ | 1.7 E$^{-2}$ | 0.9 to 1.0 E$^{-2}$ | 61 |
| 7 | Decane | 19 | 18 | 40 | 5 > | 64.7 | 0.80 | 1.3 E$^{-1}$ | 1.6 E$^{-2}$ | 0.9 to 1.2 E$^{-2}$ | 51 |
| 8 | Decane | 19 | 17 | 40 | 40 | 40.7 | 0.50 | 8.2 E$^{-2}$ | 1.7 E$^{-2}$ | 0.7 to 1.2 E$^{-2}$ | 57 |
| 9 | Dodecane | 35 | 1 | 40 | 5 > | 148.6 | 1.65 | 2.1 E$^{-1}$ | 1.5 E$^{-2}$ | 0.6 to 0.8 E$^{-2}$ | 42 |
| 10 | Dodecane | 0 | 34 | 40 | 5 > | 112.8 | 1.25 | 6.3 E$^{-2}$ | 2.5 E$^{-2}$ | 0.8 to 1.4 E$^{-2}$ | 54 |
| 11 | Dodecane | 17 | 18 | 40 | 5 > | 126.4 | 1.40 | 6.8 E$^{-2}$ | 2.4 E$^{-2}$ | 0.9 to 1.4 E$^{-2}$ | 46 |
| 12 | Dodecane | 20 | 17 | 40 | 67 | 98.8 | 1.10 | 7.2 E$^{-2}$ | 2.5 E$^{-2}$ | 0.7 to 1.1 E$^{-2}$ | 62 |

NO, NO$_2$, O$_3$, and Cl$_2$ concentrations are in ppb. RH is in %. SOA concentration is in µg m$^{-3}$. Y$_{SOA}$, $f_{44}$, $f_{57}$, and f$_{Cl}$ are dimensionless

(a) Initial concentrations

(b) Y$_{SOA}$ is calculated using the maximum SOA concentration and the initial precursor concentrations, which were 70 (octane), 81 (Decane), and 90 (Dodecane) µg m$^{-3}$, assuming complete VOC consumption

(c) Mass ratio of organic ion fragments at *m/z* 44 (presumably mostly CO$_2$$^+$) to the sum of all organic ion fragments observed at peak SOA concentration. Used as a proxy for the SOA extent of oxidation

(d) Mass ratio of organic ion fragments at *m/z* 57 (presumably C$_4$H$_9$$^+$) to the sum of all organic ion fragments observed at peak SOA concentration. Used as a proxy for hydrocarbon-like organic aerosol

(e) Ratio of particulate chlorine mass (estimated using HCl$^+$ ion fragments in the ACSM) to organic mass. Lower value is the ratio observed at organic mass peak and the higher value is the ratio observed at the particulate chlorine peak. Particulate chlorine and organics peak at different times due to the rate-limiting heterogeneous production of dihydrofurans. Chloride concentrations were near detection limits for octane experiments and may be more sensitive towards vaporizer interference effects (Wang and Hildebrandt Ruiz, 2017;Hu et al., 2017)

(f) The amount of ozone observed when peak SOA concentration was observed

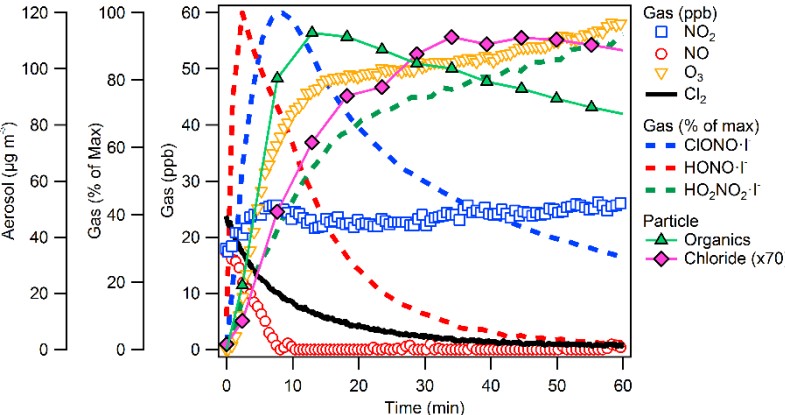

**Figure 1:** Representative trends of SOA and trace gas species during the photooxidation period. Data from dodecane oxidation (Exp. 11) are shown. Particulate chlorine concentrations were multiplied by 70 for ease of comparison. Ions consistent with HONO and $HO_2NO_2$ were observed, indicative of secondary $HO_x$ chemistry. Formation of ClONO and $ClNO_2$ due to oxidation of $NO_2$ by $Cl^{\bullet}$ was also observed.

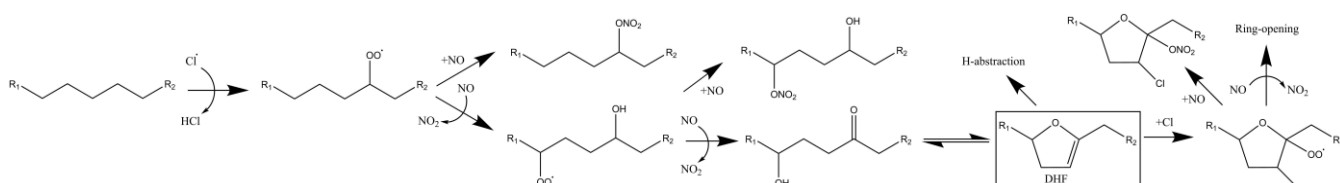

**Figure 2:** Formation pathway of chlorinated organics via chlorine addition to the DHF. H-abstraction from DHF is also possible. Ozone may also react with the double-bond on the DHF, competing with chlorine radicals.

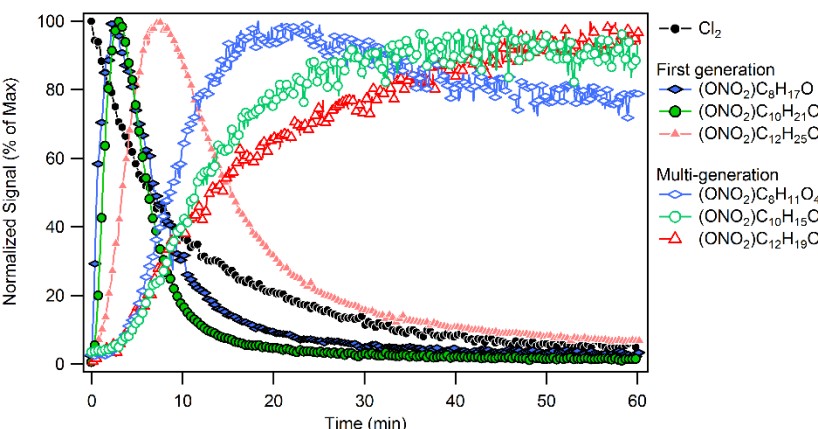

**Figure 3:** Trends of early-generation gas-phase oxidation products (hydroxynitrates) and multigeneration-oxidation products (hydroxycarbonyl nitrates) observed during chlorine-initiated oxidation of octane (Exp. 3), decane (Exp.7), and dodecane (Exp. 11). Species shown were first normalized against the $I^-$ reagent ion signal and then normalized against their respective maxima.

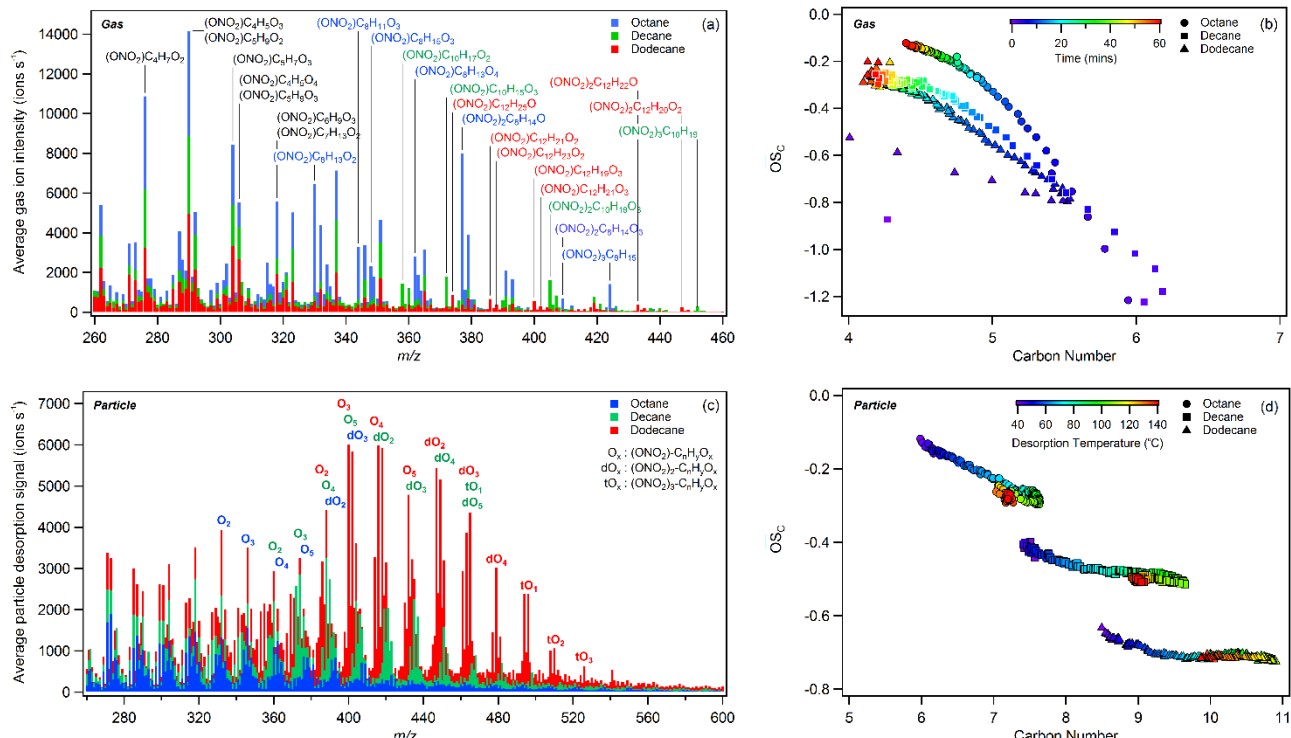

**Figure 4:** Comparison of the (a) average composition and (b) the increases in average $OS_C$ of gas phase reaction products over the photooxidation period (0-60 mins), as well as the (c) average particle-phase composition and (d) the evolution of average SOA $OS_C$ during FIGAERO desorption for octane (Exp. 3), decane (Exp. 7), and dodecane (Exp. 11). Color-coded ions in (a) and (c) indicate products that were much more abundant in reaction products derived from a particulate alkane precursor. In (c), $O_x$, $dO_x$, and $tO_x$ were used as shorthand notations for oxidized organic nitrate, dinitrate, and trinitrates with x number of oxygens (excluding those from $ONO_2$). Sticks in (a) and (c) were not stacked, and no x-axis offset was applied during plotting. In (b), $OS_C$ increased and $n_C$ decreased as photooxidation continued, consistent with photooxidative fragmentation. In (d), SOA derived from larger alkane precursors appeared less oxidized, consistent with results shown in Table 1 and Fig. S1. Methods used to calculate the average $OS_C$ and $n_C$ are detailed in the S.I.

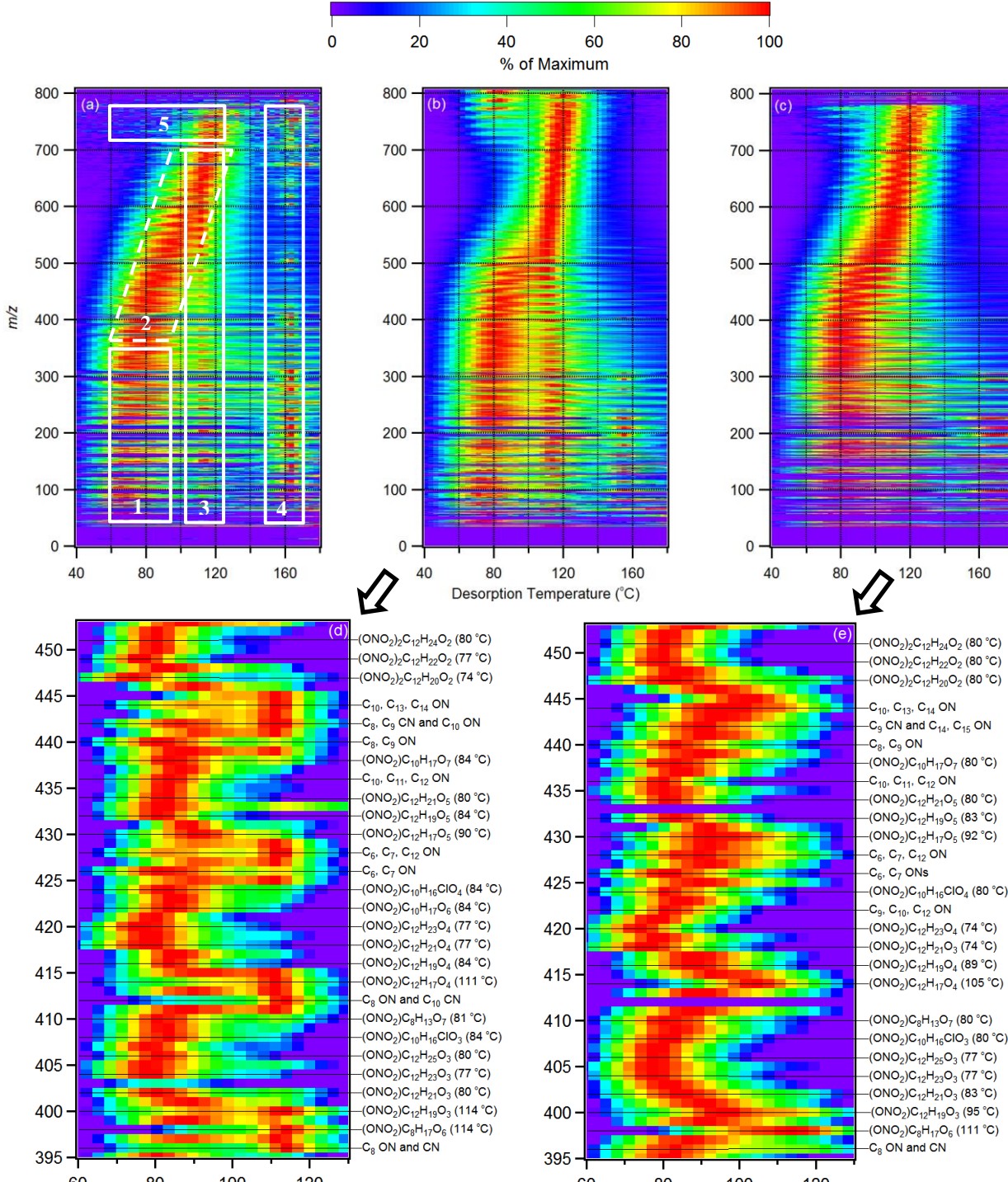

**Figure 5:** Comparison of two-dimensional thermograms for (a) octane-Cl SOA under low RH from Exp. 3, (b) dodecane-Cl SOA under low RH from Exp. 11, and (c) dodecane-Cl SOA under high RH (67 %) from Exp. 12. The color represents the ion intensity at a *m/z* as a percentage of the maximum desorption signal observed at that *m/z*. Different thermal desorption regions in (a) are dominated by (1) low-

temperature thermal decomposition products and non-adducts (2) monomers (3) oligomers and their decomposition products and (4) thermal decomposition of ammonium sulfate or extremely low volatility compounds. Region (5) is unresolved but appears to be thermally unstable oligomers. Region (2) and (3) can overlap, as shown in (d) for (b) and as shown in (e) for (c). At each nominal $m/z$, only the dominant ion is labeled in (d) and (e). If multiple ions of similar intensity were present, only general descriptions are given in the annotations, where "ON" stands for organonitrate and "CN" stands for chloronitrate. Non-nitrated organic ions were observed but not labeled because they were positioned in-between more intense organonitrates at neighboring $m/z$ coordinates. $T_{max}$ values are included in parenthesis. Increasing the RH suppressed oligomer formation, enhancing the monomer features in (c) and (e) when compared with (b) and (d). The $m/z$ values shown includes the chemical ionization reagent $I^-$ ($m/z$ 127).

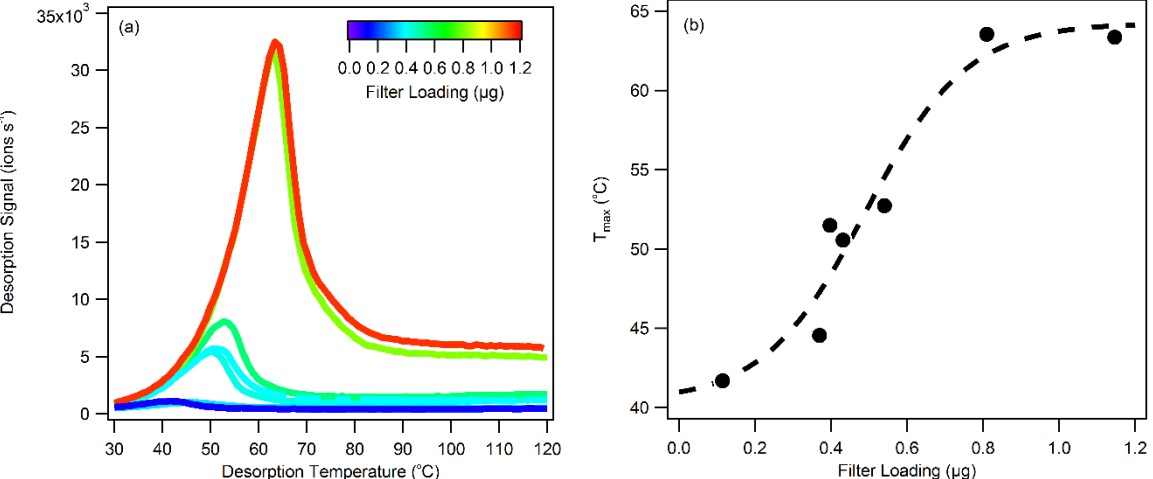

**Figure 6:** (a) The 1-D thermograms for levoglucosan ($C_6H_{10}O_5$) at different loading conditions and (b) the correlation between filter loading and $T_{max}$. Levoglucosan aerosol was generated by nebulizing a $1.2\ E^{-2}$ M aqueous solution. The aerosol was injected into the clean Teflon chamber, collected onto the FIGAERO filter, and analyzed. The $T_{max}$-loading correlation for pure levoglucosan could be described by a sigmoid function, leveling off at 41 and 64 $^\circ$C under very low and very high loading conditions, respectively.