# Peer review of "Chlorine-initiated oxidation of n-alkanes under high NOx conditions: Insights into secondary organic aerosol composition and volatility using a FIGAERO-CIMS"

_Atmospheric Chemistry and Physics, 2018_

## Referee Comment (RC1) · Anonymous Referee #1 · 10 Jul 2018

General Comments

This paper reports results of a laboratory study of secondary organic aerosol (SOA) formation from the oxidation of a series of n-alkanes with Cl atoms in the presence of NOx in a smog chamber. The gas-phase composition of products and SOA was analyzed using a FIGAERO-CIMS to obtain information on product molecular formulas and thermal desorption profiles, and an ACMS and SEMS were used as well. The

results were then interpreted in terms of the reasonably well-established mechanism for alkane oxidation by OH radicals, with the addition of possible oligomer-forming reactions. Interesting observations are reported on chlorine-containing products, organic nitrates, other multi-generation oxygenated products, and a new 2-D format is used to present composition and thermal data. The results provide a nice comparison to OH oxidation and the molecular and thermal characterization of products may be useful for future development and testing of models for these reactions. The experiments are very well done, and the introduction, data analysis, and interpretation are thorough and clearly written. Cl oxidation of VOCs has not been well studied, and as shown here its products can differ in significant ways from OH oxidation. There is thus plenty more to be done, but this paper is a good starting point for detailed SOA chemistry studies. I think the paper can be published in ACP after the following comments are addressed.

Specific Comments

1. In a study of the aging of pentadecane by OH/NOx that was not cited here (Aimanant and Ziemann, AS&T, 2013), evidence was presented based on thermal desorption profiles for the presence of acetal dimers formed from cyclic hemiacetals. These would likely be very stable and so probably desorb intact. Do the authors see any evidence for such oligomers?

2. Can the composition data be used to estimate average molecular weights for the SOA analyzed by the FIGAERO-CIMS? It would be useful to use these values to convert the mass yields of SOA to molar yields of SOA, since this gives a better indication of the efficiency of SOA formation. Otherwise the large mass yields (due in large part to the extensive oxidation) give the impression that a large fraction of the alkane is converted to SOA, which is not really true.

3. Is there any indication for chemistry occurring between Cl, O2, and NOx? Would this be expected?

4. Structure-activity relationships for the reactions of alkanes with Cl atoms indicate

that the ratio of rate constants for abstraction of primary/secondary H-atoms is significantly smaller than for OH reactions, indicating that more reaction occurs at the ends of the alkanes with Cl than with OH. Have the authors considered what the consequences of this might be with regards to the products being formed?

5. Have the authors considered comparing vapor pressures estimated from the measured desorption temperatures with those predicted from proposed molecular structures and the SIMPOL group contribution method?

6. It would be useful to provide a little more detailed speculation on the proposed nature of the thermal decomposition products. Are these thought to be formed by simple reversible dissociation of hemiacetals, for instance, or by cleavage of strong C-C or C-O bonds? Or might they just be compounds that are "trapped" in a solid SOA matrix that cannot escape until the SOA melts?

7. I did not see a quoted density for calculating SOA yields from SEMS data. This value and its source should be noted.

Technical Comments

None.

---

## Referee Comment (RC2) · Anonymous Referee #2 · 17 Jul 2018

General comments

Wang et al. present results from environmental chamber experiments of C8-C12 alkane oxidation initiated by chlorine under dry/humid and high NOx conditions. They show that yields are higher than for OH-initiated oxidation of the respective alkanes, indicative of the importance of these reactions for SOA production. Identified compounds include organonitrates, organochlorides. FIGAERO-CIMS data are used to look into
the connection of chemistry and volatility via a new way of representing thermograms.

Overall, this is a very well written paper, and the study and interpretation of results are sound. I therefore recommend this paper to be published after minor revisions. Apart from a few specific comments (see below), I have two general comments regarding the manuscript.

First, the manuscript would profit from a few more lines on its atmospheric relevance. Using alkanes as VOC precursors, and performing experiments under high NOx conditions indicates the authors wanted to simulate an urban/polluted atmosphere. A short discussion on this, including the importance of Cl- oxidation in such environments, as well as the choice of RH conditions, would give the study more (atmospheric) importance.

Second, there is somewhat a disconnect in the narrative between section 3.1 and section 3.2, which also represents a disconnect between ACSM and FIGAERO-CIMS data. I suggest the authors try to connect these two parts better. Section 3.1 (SOA and organic chloride formation) is entirely ACSM based. Why? Why were FIGAERO-CIMS data not used? Some statements made in a tentative manner could be confirmed/looked into using FIGAERO-CIMS data (see specific comments below).

Specific comments

P. 1, l. 26/ p. 2 l. 11: Where is chlorine oxidation important? See comment above, the paper would benefit from a few more lines on its atmospheric relevance. P. 2, l. 32 – 33: You specifically mention here low NOx, as presumably the cited study was done under such. What are then the implications for your study? I suggest reformulating this sentence. P. 4, l. 28: UMR, why not HR? In Figure 5 you present molecular formulae of compounds, indicating HR analysis. Please clarify. P. 5, l. 11: This could be confirmed with FIGAERO-CIMS data. Why were they not added? P. 5, l. 18 – 19: I- should cluster with Cl2. Why do you use the Cl- signal to track Cl2? Please clarify. P. 5, l. 21 – 22: Did you see any evidence of that in FIGAERO-CIMS gas-phase measurements? P. 6,

l. 1-2: Can you confirm that with the organonitrate measurements of the FIGAERO-CIMS? P. 6, l. 16 – 18: Also here the isotope signal should help you. Without that, the chloronitrate peak cannot be identified (based on Figure S8). P. 6, l. 21 – 22: Did you actually observe that as well if you compare your dry and humid experiments, no? You mention this as a finding in your conclusions as well, but I cannot find it as a result in the manuscript. P. 7, l. 3-4: Are those fragmentation reactions in the particle phase, with subsequent evaporation of the resulting compounds? This would be consistent with the observation of loss of SOA mass (mentioned e.g. on p. 5, l. 21 – 22). Please elaborate. Experiments shown here are all under dry conditions. What about humid conditions? P. 7, l. 13 – 14: It becomes clear after discussion of Figure 5, but it would be helpful for the reader to mention here why you use the temperature range of $40 - 140$ °C. P. 8, l. 8 – 10: What do you base your interpretation of "low-temperature thermal fragmentation products" on? I agree that there must be fragmentation, but I am not sure you have enough evidence for that being thermal fragmentation during desorption. P. 9, section 3.3: How reproducible are your thermograms and corresponding Tmax for one compound and stable conditions? This information should be added e.g. to the supplementary section. P. 11, l. 22 – 23: This sentence is formulated too strongly based on the observations you present in your results section.

Technical corrections

P. 6, l. 8: Should be ACSM P. 8, l. 3: Propose

---

## Author Comment (AC1) · 29 Aug 2018

Author Response to Comments by Anonymous Referees

We thank the referees for their suggestions and recommendations. Below are our responses to all comments.

(RC): Referee comment
(AR): Author response

Anonymous Referee #1

*This paper reports results of a laboratory study of secondary organic aerosol (SOA) formation from the oxidation of a series of n-alkanes with Cl atoms in the presence of NOx in a smog chamber. The gas-phase composition of products and SOA was analyzed using a FIGAERO-CIMS to obtain information on product molecular formulas and thermal desorption profiles, and an ACMS and SEMS were used as well. The results were then interpreted in terms of the reasonably well-established mechanism for alkane oxidation by OH radicals, with the addition of possible oligomer-forming reactions. Interesting observations are reported on chlorine-containing products, organic nitrates, other multi-generation oxygenated products, and a new 2-D format is used to present composition and thermal data. The results provide a nice comparison to OH oxidation and the molecular and thermal characterization of products may be useful for future development and testing of models for these reactions. The experiments are very well done, and the introduction, data analysis, and interpretation are thorough and clearly written. Cl oxidation of VOCs has not been well studied, and as shown here its products can differ in significant ways from OH oxidation. There is thus plenty more to be done, but this paper is a good starting point for detailed SOA chemistry studies. I think the paper can be published in ACP after the following comments are addressed.*

We thank the reviewer for the detailed assessment of the manuscript. As suggested and detailed below, we have expanded the discussions on both the gas-phase oxidation chemistry and particle-phase composition.

*(1, RC) In a study of the aging of pentadecane by OH/NOx that was not cited here (Aimanant and Ziemann, AS&T, 2013), evidence was presented based on thermal desorption profiles for the presence of acetal dimers formed from cyclic hemiacetals. These would likely be very stable and so probably desorb intact. Do the authors see any evidence for such oligomers?*

(1, AR): Ions consistent with acetal and hemiacetal dimers as proposed by Aimanant and Ziemann (2013) were identified in the SOA using the FIGAERO-CIMS. In addition, ions consistent with hemiacetal dimers formed via reaction between hydroxycarbonyl nitrate and hydroxy dihydrofuran as proposed by Schilling Fahnestock et al. (2015) were identified. A diagram depicting the potential formation mechanisms, Fig. S10, is added to the S.I. A discussion on oligomerization pathways and identification of oligomeric compounds is added to the main text. Bimodal thermal desorption behaviors observed for numerous ions indicate that some oligomers underwent thermal decomposition. It is difficult to estimate the fraction of oligomer that desorbed intact, given how the decomposition mode (second desorption signal peak in a thermogram) for many ions share the same $T_{max}$, as illustrated in Figs. 5 and S7.

**Addition of S.I. figure:**

**Figure S10:** Oligomerization via the reaction between (a) carbonyl hydroxyl nitrate and hydroxy dihydrofuran proposed by Schilling Fahnestock et al. (2015) and (b) cyclic hemiacetal with either a 1,4-hydroxycarbonyl or another cyclic hemiacetal proposed by Aimanant and Ziemann (2013). Condensed-phase isomerization between acetal and hemiacetal dimers is also possible, as shown in (b).

**Addition to Results and Discussion:** "Oligomerization may result from the condensed-phase reaction of two cyclic hemiacetal compounds, which are expected to be in the particle phase, to form a acetal dimer or from the reaction between a cyclic hemiacetal and a 1,4-hydroxycaronbyl compound, which may be present in the gas or particle phase, to form a hemiacetal (Aimanant and Ziemann, 2013). Condensed-phase, acid-catalyzed isomerization of hemiacetal and acetal is also possible (Aimanant and Ziemann, 2013). For OH-initiated oxidation of dodecane under high $NO_x$ conditions, hemiacetals and nitrate hemiacetals are reported to dominate the oligomer composition (Schilling Fahnestock et al., 2015), where nitrate hemiacetals could form via the reactive uptake of hydroxy dihydrofuran and hydroxycarbonyl nitrate compounds (Schilling Fahnestock et al., 2015). The three oligomerization pathways are illustrated in Fig. S10. Ions consistent with acetal, hemiacetal, and nitrate hemiacetal compounds were observed in the particle phase using the FIGAERO-CIMS. Oligomer formation via aldol condensation by highly oxidized compounds is also possible (Schilling Fahnestock et al., 2015). It should be noted that the oligomerization may also proceed through a $NO_3$-elimination, reverse esterification process involving two alkyl nitrate compounds to form a singly nitrate oligomer, which has been recently proposed for α-Pinene, though the exact mechanism remain elusive (Faxon et al., 2017)."

*(2, RC): Can the composition data be used to estimate average molecular weights for the SOA analyzed by the FIGAERO-CIMS? It would be useful to use these values to convert the mass yields of SOA to molar yields of SOA, since this gives a better indication of the efficiency of SOA formation. Otherwise the large mass yields (due in large part to the extensive oxidation) give the impression that a large fraction of the alkane is converted to SOA, which is not really true.*

(2, AR): Based on FIGAERO-CIMS data, assuming equal sensitivity for all ions, the average molecular weight, $mw_{avg}$ is estimated based on the desorption intensity $I_i$ and the molecular weight, $mw_i$ of all ions identified,

$$mw_{avg}(T_d) = \frac{\sum_i mw_i \times I_i(T_d)}{\sum_i I_i(T_d)}$$

where $mw_{avg}$ and $I_i$ vary with desorption temperature, $T_d$ during a single FIGAERO desorption. A single average molecular weight for the entire FIGAERO desorption run can be calculated based on the integrated values over the $T_d$ range,

$$mw_{SOA} = \frac{\int (\sum_i mw_i \times I_i(T_d)) \Delta T_d}{\int (\sum_i I_i(T_d)) \Delta T_d}$$

where $\Delta T_d$ is the step change in desorption temperature. It should be noted that because of thermal decomposition, which has been observed for monomers as well (Stark et al., 2017), the $mw_{avg}$ and $mw_{SOA}$ calculated using the FIGAERO-CIMS data likely underestimate the actual average SOA molecular weight. For octane SOA, $mw_{SOA}$ calculated in this way ranges from 206 to 226 g mol$^{-1}$; for decane SOA, $mw_{SOA}$ ranges from 241 to 265 g mol$^{-1}$; for dodecane SOA, $mw_{SOA}$ ranges from 260 to 270 g mol$^{-1}$. The molar SOA yield can then be calculated as,

$$Y_{molar} = \frac{n_{SOA}}{\Delta n_{VOC}} = \frac{m_{SOA}/mw_{SOA}}{\Delta m_{VOC}/mw_{VOC}} = Y \frac{mw_{VOC}}{mw_{SOA}}$$

where $n_{SOA}$ and $n_{VOC}$ are the molar concentrations of SOA and VOC. The molecular weight of VOC, $mw_{VOC}$ is known for the alkane precursors. The corresponding molar yield ranges from 0.08 to 0.14 for octane SOA, 0.26 to 0.47 for decane SOA, and 0.72 to 1.04 for dodecane SOA. The above-unity molar SOA yield observed for dodecane (Exp. 9) indicates that SOA mass may be overestimated, which may be the result of uncertainties with the collection efficiency (CE) and relative ionization efficiency (RIE) assumed for the SOA, which are addressed by (7. AR) below in response to (7. RC). The method described above is added to the S.I..

*(3, RC): Is there any indication for chemistry occurring between Cl, O2, and NOx? Would this be expected?*

(3, AR): Ions consistent with Cl and/or NO$_x$ oxidation products, including HNO$_3$, ClONO (or ClNO$_2$), ClO, HONO, HO$_2$NO$_2$, and N$_2$O$_5$ were observed with the I$^-$ CIMS, which are consistent with literature findings. Selected time-series are added to Fig. 1 in the main text and to Fig. S# in the S.I.. The results suggest that secondary HO$_x$ chemistry and perhaps NO$_3^-$ chemistry (in the dark) may be important, which warrant future investigations. Discussion of the interactions between Cl, O$_3$, and NO$_x$ are added to the main text.

**Updates to Fig. 1:**

[Figure]

**Fig 1.** Representative trends of SOA and trace gas species during the photooxidation period. Data from dodecane oxidation (Exp. 11) are shown. Particulate chlorine concentrations were multiplied by 70 for ease of comparison. Observation of select ions such as HONO and $HO_2NO_2$ are indicative of secondary $HO_x$ chemistry. Formation of ClONO from the interaction between Cl$^\bullet$ and $NO_2$ was also observed.

**Addition of Fig. S11:**

[Figure]

**Fig. S11:** Representative trends of SOA and trace gas species during the photooxidation period. Data from dodecane oxidation (Exp. 11) are shown, similar to Fig. 1 in the main text. Additional species shown include $ClONO_2$ (from reaction between ClONO with $NO_2$, Lesar et al., 2006), ClO (possibly a ion fragment of ClONO), HCl (from H-abstraction by Cl$^\bullet$ or perhaps Cl-elimination reactions), and $HNO_3$ (due to the oxidation of $NO_2$ by OH).

**Addition to Results and Discussion:** "Ions consistent with $HO_2NO_2$, HONO, and $ClNO_2$ were observed in the gas-phase as well, as shown in Fig. 1. Formation of $HO_2NO_2$ from the reaction between $HO_2$ and $NO_2$, as well as the detection capability of I⁻ CIMS towards it have been reported in the literature (Veres et al., 2015),

$$HO_2 + NO_2 \xrightarrow{M} HO_2NO_2 \qquad\qquad \text{Eq. (2)}$$

where HO$_2$ may be produced from the Cl-initiated oxidation of alkanes. The presence of HO$_x$ radical also enables the formation of HONO under UV,

$$HO_2 + NO \rightarrow OH + NO_2 \qquad \text{Eq. (3)}$$
$$OH + NO \rightarrow HONO \qquad \text{Eq. (4)}$$

which, as with HO$_2$NO$_2$ formation, is a strong indication of secondary HO$_x$ chemistry enabled by primary Cl radical chemistry. As expected, HONO readily decays under UV to background levels over the course of the photooxidation. Ions consistent with nitryl chloride (ClNO$_2$) are expected to correspond primarily to chlorine nitrite (ClONO): The reaction between Cl$^{\bullet}$ and NO$_2$ forms both ClONO (> 80 % of the time) and ClNO$_2$ (< 20 % of the time) as reaction products (Golden, 2007; Niki et al., 1978)."

*(4, RC): Structure-activity relationships for the reactions of alkanes with Cl atoms indicate that the ratio of rate constants for abstraction of primary/secondary H-atoms is significantly smaller than for OH reactions, indicating that more reaction occurs at the ends of the alkanes with Cl than with OH. Have the authors considered what the consequences of this might be with regards to the products being formed?*

(4, AR): With increased terminal H-abstraction by Cl radicals as compared to OH-radicals, we would expect increased primary alkyl nitrate formation. This should result in lowered saturation vapor pressure of the oxidation products (Lim and Ziemann, 2009b; Yeh and Ziemann, 2014, 2015), consistent with the higher SOA yield observed here. We have added a discussion of this factor to the main text.

**Addition to Introduction:** "According to the structure-activity relationship, the abstraction of terminal hydrogen occurs more frequently with alkane-Cl than with alkane-OH reactions (Kwok and Atkinson, 1995). This results in different product distributions; for example, increased formation of primary alkyl nitrates."

**Addition to Results and Discussion**: "Formation of primary alkyl nitrates, which have lower saturation vapor pressures than the secondary alkyl nitrates, due to terminal H-abstraction would also contribute to the higher SOA yields observed for alkane-Cl than for alkane-OH oxidation"

*(5, RC): Have the authors considered comparing vapor pressures estimated from the measured desorption temperatures with those predicted from proposed molecular structures and the SIMPOL group contribution method?*

(5, AR): Previous FIGAERO-CIMS studies show that vapor pressures calculated using either the C$^*$-T$_{max}$ correlation or the direct partitioning method (Lopez-Hilfiker et al., 2015) can have significant disagreements with SIMPOL results (D'Ambro et al., 2017; Stark et al., 2017), which have been mostly attributed to the effects of thermal decomposition. Another complication with C$^*$ estimation can also arise from potential filter loading dependence for T$_{max}$ (Huang et al., 2018 and this work). In our opinion, this requires further investigation before vapor pressure measurements by FIGAERO and SIMPOL may be reconciled.

*(6, RC): It would be useful to provide a little more detailed speculation on the proposed nature of the thermal decomposition products. Are these thought to be formed by simple reversible dissociation of hemiacetals, for instance, or by cleavage of strong C-C or C-O bonds? Or might they just be compounds that are "trapped" in a solid SOA matrix that cannot escape until the SOA melts?*

(6, AR): Given that numerous organic ions exhibit bimodal thermal desorption behavior, it is reasonable to attribute it to the reversible dissociation of oligomers, a phenomenon that has been observed in the FIGAERO at room temperatures (Schobesberger et al., 2018) and at high desorption temperatures (Faxon et al., 2017). Decomposition of pure monomeric organic acids has also been reported previously, indicating that cleavage of C-C and C-O bonds (C-OH or C-OOH) does occur in the FIGAERO (Stark et al., 2017). Trapping of molecular compounds inside the SOA matrix may be possible if the aerosol existed in a "glassy state", and could result in a sudden release of volatilized vapors (all having similar $T_{max}$, as seen in region 1 and 3) though the aerosol phase state cannot be evaluated based on data available. It is worth noting that ions associated with ammonium sulfate decomposition all share the same $T_{max}$ with or without SOA present (that is, uniform $T_{max}$ can be the result of thermal decomposition as well).

**Addition to Results and Discussion**: "The thermal decomposition products may be the result of reversible dissociation of oligomers (Faxon et al., 2017; Schobesberger et al., 2018) or the cleavage of C-O or C-C bonds, which has been observed in the thermal decomposition of monomeric organic acids (Stark et al., 2017). Additionally, SOA produced under dry conditions may exist in a "glassy" state (Koop et al., 2011), trapping volatile components (Perraud et al., 2012) that are released as the SOA mass is depleted.

*(7, RC): I did not see a quoted density for calculating SOA yields from SEMS data. This value and its source should be noted.*

(7, AR): The SOA yield was calculated from the ACSM measurement of organic aerosol concentration, assuming a collection efficiency, CE of 0.5 and a relative ionization efficiency, RIE of 1.4. The average, apparent SOA density calculated using ACSM and SEMS data is around $2.1 \pm 0.3$ g cm$^{-3}$, which is higher than the densities previously reported for OH-alkane SOA formed under high NO$_x$ conditions, 1.06 to 1.28 g cm$^{-3}$ (Lim and Ziemann, 2009a; Loza et al., 2014). The discrepancy can result from the differences in SOA composition, aerosol oxidation state (Kuwata et al., 2012), or from instrument uncertainties. For instance, studies report that RIE of organics may depend on the aerosol oxidation state of carbon (Li et al., 2018; Xu et al., 2018). For $-1 < OS_C < 0.5$, which overlaps with the average $OS_C$ as indicated by the FIGAERO-CISM measurement as shown in in Fig. 4d, the average RIE is reportedly $1.6 \pm 0.5$ instead of 1.4 (Xu et al., 2018), which is higher than the RIE value of 1.4 assumed here. Re-calculation using the updated RIE value would lower the apparent SOA density closer to agreement with previous studies. Without an independent measurement for particle mass (e.g. using a particle mass analyzer), it is not possible to ascertain the exact SOA density. Additional discussion associated with SOA quantification are added to the main text.

**Additions to Methods:** "Aerosol concentration was calculated using ACSM measurement assuming a collection efficiency of 0.5 and RIE of 1.4, corrected for depositional wall loss […]"

**Additional to Results of Discussion:** "The apparent SOA density, calculated using the ACSM SOA mass measurement the SEMS SOA volume measurement, was $2.1 \pm 0.3$ g cm$^{-3}$, which is substantially higher than that reported for OH-alkane SOA formed under high NO$_x$ conditions, 1.06 to 1.28 g cm$^{-3}$ (Lim and Ziemann, 2009a; Loza et al., 2014). The differences could be due to uncertainties associated with the ACSM and similar instruments, specifically that of the RIE (Li et al., 2018; Xu et al., 2018) and CE (Docherty et al., 2013; Middlebrook et al., 2012; Robinson et al., 2017), which may vary with SOA composition, oxidation state, or phase-state. Assuming a SOA density of 1.06 g cm$^{-3}$, the SOA yield (0.08 to 0.88) is still higher for Cl-alkane oxidation as compared to OH-alkane oxidation."

(AR): Lines specified by the referee is copied below from the ACPD manuscript; revisions are noted in the author response

*(1, RC) Wang et al. present results from environmental chamber experiments of C8-C12 alkane oxidation initiated by chlorine under dry/humid and high NOx conditions. They show that yields are higher than for OH-initiated oxidation of the respective alkanes, indicative of the importance of these reactions for SOA production. Identified compounds include organonitrates, organochlorides. FIGAERO-CIMS data are used to look into the connection of chemistry and volatility via a new way of representing thermograms. Overall, this is a very well written paper, and the study and interpretation of results are sound. I therefore recommend this paper to be published after minor revisions. Apart from a few specific comments (see below), I have two general comments regarding the manuscript.*

*First, the manuscript would profit from a few more lines on its atmospheric relevance. Using alkanes as VOC precursors, and performing experiments under high NOx conditions indicates the authors wanted to simulate an urban/polluted atmosphere. A short discussion on this, including the importance of Cl- oxidation in such environments, as well as the choice of RH conditions, would give the study more (atmospheric) importance.*

(1, AR): We thank the referee for the recommendations. We have added some discussions on the experimental design choices and the atmospheric relevance of Cl oxidation chemistry

**Revisions / Additions to Introduction**: "ClNO$_2$ photolysis in the early morning produces Cl radical and NO$_x$, which has been shown to enhance RO$_2$ production from alkane oxidation in near coastal regions (Riedel et al., 2012) as well as OH radical propagation in urban environments (Young et al., 2014). In addition to reactive chlorine emissions from water treatment (Chang et al., 2001) and fuel combustion (Osthoff et al., 2008; Parrish et al., 2009), the rising usage of volatile chemical products (VCP) such as pesticides, cleaning products, and personal care products can be a significant sources for reactive chlorine compounds and VOCs in urban environments (Khare and Gentner, 2018; McDonald et al., 2018)" [..] "Cl-VOC oxidation products such as isomers 1-chloro-3-methyl-3-butene-2- (CMBO), a tracer for isoprene-chlorine chemistry (Nordmeyer et al., 1997), have been observed in highly polluted environments (Le Breton et al., 2018; Tanaka et al., 2003)" […] "Observation of isoprene-derived organochloride was recently reported for filter samples collected in Beijing (Le Breton et al., 2018)." […]

**Addition to Methods**: "Under typical atmospheric conditions, elevated RH can be expected, especially within the marine boundary layer in near-coastal regions, where Cl-alkane chemistry may be important (Riedel et al., 2012). Therefore, SOA formation under humid conditions were also investigated.

**Addition to Conclusion**: "Overall, these results show that chlorine-alkane oxidation could be an important pathway for SOA production and ageing, especially in highly polluted environments replete with alkane, NO$_x$, and reactive chlorine emission sources."

*(2, RC): Second, there is somewhat a disconnect in the narrative between section 3.1 and section 3.2, which also represents a disconnect between ACSM and FIGAERO-CIMS data. I suggest the authors try to connect these two parts better. Section 3.1 (SOA and organic chloride formation) is entirely ACSM based. Why? Why were FIGAEROCIMS data not used? Some statements made in a tentative manner could be confirmed/ looked into using FIGAERO-CIMS data (see specific comments below).*

(2, AR): The revised manuscript makes more use of quantitative FIGAERO-CIMS data, and compares ACSM to FIGAERO data. Additional gas- and particle-phase results are presented in the response to the following questions.

*(3, RC): P. 1, l. 26/ p. 2 l. 11: Where is chlorine oxidation important? See comment above, the paper would benefit from a few more lines on its atmospheric relevance.*

P.1 l.26: "Evaporation of POA due to dilution can provide additional gas-phase alkanes, which can undergo photooxidation initiated by OH, NO3, as well as chlorine radicals (Aschmann and Atkinson, 1995; Atkinson and Arey, 2003)."

P.2 L. 11: "Recent field studies have identified reactive chlorine compounds in diverse locales from natural and anthropogenic sources (Faxon and Allen, 2013; Finlayson-Pitts, 2010; Saiz-Lopez and von Glasow, 2012; Simpson et al., 2015)."

(3, AR): We have expanded the discussion on the conditions under which chlorine chemistry may be important as suggested as detailed in (1, AR).

*(4, RC): P. 2, l. 32 –33: You specifically mention here low NOx, as presumably the cited study was done under such. What are then the implications for your study? I suggest reformulating this sentence.*

P.2. L.32-33: "Chlorine radicals can react with dihydrofuran via both H-abstraction and Cl-addition, producing chlorinated (e.g. dichlorotetrahydrofurans) and non-chlorinated compounds (e.g. furanones) under low NOx conditions (Alwe et al., 2013)."

(4, AR): The referenced study was performed under low NOx conditions. The main implication here is that the reaction between Cl and dihydrofuran can produce both chlorinated and non-chlorinated products. Under high NOx conditions, both chlorinated and non-chlorinated organonitrate compounds may be formed. The following sentence has been added to clarify the implication of the NOx study and its connection with the high NOx scenario.

**Addition to Introduction:** "…under low NOx condition (Alwe et al. 2013). Similarly, formation of both chloronitrates (via Cl-addition) and organonitrates (via H-abstraction) from alkane-Chl oxidation is possible in the presence of NOx."

*(5, RC): P. 4, l. 28: UMR, why not HR? In Figure 5 you present molecular formulae of compounds, indicating HR analysis. Please clarify.*

P.4 L.28: "The 2-D thermogram is comprised of normalized unit-mass resolution 1-D thermograms, each expressed as a percentage color scale of the maximum desorption signal."

(5, AR): We have conducted HR analysis and have experimented with a HR 2-D thermogram as the referee alluded to. We decided to present the UMR version due to its simplicity, where the multimodal thermal desorption behavior and a general $T_{max}$-molecular weight relationship can be visualized in the absence of explicit molecular formulae, which is especially helpful for studying the desorption behavior of high molecular weight compounds which may have a vast number of possible molecular makeups. Another practical concern with the HR 2-D thermogram is the uneven spacing of the molecular weight of the identified ions, which presents some technical issues for plotting on a heat map. We have added a brief discussion on the advantages and disadvantages of using UMR vs HR data for constructing the 2-D thermogram.

**Change / Addition to Methods**: "The 2-D thermogram is comprised of normalized unit-mass resolution (UMR) 1-D thermograms, each expressed as a percentage color scale of the maximum desorption signal. The application of the 2-D thermogram is described in Section 3.2. The advantage of using UMR over HR data is the ability to investigate the SOA thermal desorption behavior over the entire m/z and volatility (i.e. $T_{max}$) space without having to explicitly assign chemical formulae to individual ions, which can be a time-consuming process, especially for high molecular weight compounds whose exact molecular composition may be difficult to ascertain. The disadvantage of using UMR over HR data is the potential interference by isotopic signals or non-adduct ions. Therefore, the 2-D thermogram should be used as compliment rather than a replacement to the HR analysis."

*(6, RC): P. 5, l. 11: This could be confirmed with FIGAERO-CIMS data. Why were they not added?*

P.5, L. 11: "Given the same oxidation conditions, SOA products derived from longer alkane precursors appeared less oxidized…"

(6, AR): We have added Fig. 4(d) to support this observation using the FIGAERO-CIMS data, which shows that the SOA oxidation state decreases with increasing precursor length in agreement with the ACSM data analysis as shown in Table 1 and Fig. S1. Figure 4(b) in the ACPD manuscript is now Fig. 4(c).

**Addition to the S.I.**: Using the oxygen-to-carbon ratio (O:C) as the example, the bulk SOA elemental ratios as a function of the FIGAERO desorption temperature, $T_d$ can be calculated as

$$O:C_{avg}(T_d) = \frac{n_O(T_d)}{n_C(T_d)} == \frac{\sum_i n_{O,i} \times I_i(T_d)}{\sum_i n_{C,i} \times I_i(T_d)}$$

where $n_O$ is the total number of oxygen atoms present in the desorbed ions at $T_d$, $n_C$ is the total number of carbon atoms, $n_{O,i}$ is the number of oxygen atom present in compound *i* as determined by its assigned molecular formula (which is independent of $T_d$), $n_{C,i}$ is the number of carbon present in compound *i*, and $I_i$ is the desorption ion intensity for compound *i* at $T_d$. Equal

sensitivity is assume for ions used for the analysis such that $I_i$ can be used as the molar amount for compound *i*.

**Addition to Methods:** "The average oxidation state of carbon ($OS_C$) may be estimated using parameterization based on $f_{44}$ and $f_{43}$ as measured by the ACSM, which are themselves proxies for relative contributions of more oxidized and less oxidized organic compounds to the SOA mass (Ng et al., 2011). $OS_C$ may also be calculated based on the SOA molecular composition as observed by the FIGAERO-CIMS, where

$$OS_C = 2 \times O{:}C - H{:}C + NO_3{:}C + Cl{:}C \qquad \text{Eq. (1)}$$

where $NO_3{:}C$, $Cl{:}C$, $O{:}C$, and $H{:}C$ are the molecular ratios of the number of -$NO_3$ functional group, -Cl functional group, non-$NO_3$ oxygen atoms, and H atoms to the number of carbon atoms for any given compound. The average SOA $OS_C$ is calculated based on iodide-adduct only, assuming equal sensitivity for all organic compounds"

**Addition of Fig. 4(d):**

[Figure]

**Figure 4d.** Average oxidation state of carbon ($OS_C$) and carbon number of ions observed during the temperature-programmed desorption of SOA produced from the Cl-initiated oxidation of octane (Exp. 3), decane (Exp. 7), and dodecane (Exp. 11). Under similar oxidant conditions, SOA derived from longer VOC precursor were on average less oxidized.

*(7, RC): P. 5, l. 18 – 19: I⁻ should cluster with Cl₂. Why do you use the Cl⁻ signal to track Cl₂?*

P.5, L.18-19: The $Cl_2$ concentration was estimated using I⁻ CIMS by tracking the Cl⁻ ion.

(7, AR): Cl⁻ ion was used at one point during the drafting of the manuscript. The $Cl_2$I⁻ ion was ultimately used in Fig. 1 shown in the discussion paper but the text was not updated by mistake. This has been corrected.

Revision: "The Cl₂ concentration was estimated using I⁻ CIMS by tracking the Cl₂I⁻ ion"

*(8, RC): Please clarify. P. 5, l. 21 – 22: Did you see any evidence of that in FIGAERO-CIMS gas-phase measurements?*

P. 5, L. 21-22: "Oxidation continued under UV driven by chlorine radicals, and the SOA concentration began to decay due to oxidative fragmentation (Kroll et al., 2011; Lambe et al., 2012; Wang and Hildebrandt Ruiz, 2017)."

(8, AR): We have added Fig. 4(b) to show the simultaneous increases in the average $OS_C$ and decreases in the $n_C$ of gas-phase species as observed the I⁻ CIMS over the course of the photooxidation (0 to 60 mins), which are consistent with oxidative fragmentation. Results also show that octane oxidation products are significantly more oxidized compared to decane and dodecane oxidation products.

**Addition of Fig. 4(b):**

[Figure]

Figure 4(b). Average $OS_C$ and $n_C$ of gas-phase compounds observed during the photooxidation period for Exp. 3 (Octane), 7 (decane), and 11 (dodecane). As oxidation continues, $OS_C$ increased and $n_C$ decreased, consistent with oxidative fragmentation.

*(9, RC): P. 6, l. 1-2: Can you confirm that with the organonitrate measurements of the FIGAEROCIMS?*

P.6, L.1-2: "Higher initial NO concentrations led to higher SOA yields and lower ozone production for all precursors, as shown in Table 1 and Fig. S2. This is similar to alkane OH SOA formation, where higher NO concentrations lead to more abundant organic nitrate formation, which increases the SOA volume (Schilling Fahnestock et al., 2015) and density (Loza et al., 2014)."

(9. AR): FIGAERO data do show increased organonitrate abundance in the particle phase for SOA formed under NO-only conditions as compared to NO₂-only conditions in terms of both

molar ratio (i.e. average number of -NO$_3$ functional group per molecule) and mass fraction (i.e. mass of organonitrate as a fraction of the identified SOA mass). For mixed NO-NO$_2$ conditions, the observed organonitrate molar ratio and mass fraction are closer to that observed under the NO$_2$ only condition. The results are incorporated into the main text. The tabulated data are added to the S.I.

**Addition to Results and Discussion:** "[…] FIGAERO-CIMS shows that the particulate organonitrate molar ratio (calculated as the average number of -NO$_3$ per molecule) and mass fraction were the highest in SOA produced under NO-only conditions (see Table S1). The organonitrate molar ratio also increased with the alkane precursor length, from 0.57-0.64 for octane to 0.75-0.93 for dodecane. Similar trend was observed for the orgnaonitrate mass fraction, which increased from 0.53-0.58 for octane to 0.66-0.72 for dodecane."

*(10, RC): P. 6, l. 16 – 18: Also here the isotope signal should help you. Without that, the chloronitrate peak cannot be identified (based on Figure S8).*

P. 6, L. 16-18: "Compounds resembling chloronitrates (e.g. ONO2 C12H18ClO2•I- for dodecane) were tentatively identified in the particle-phase using the FIGAERO but they were not well separated from the shoulder of nearby organonitrate peaks (e.g. ONO2 C12H21O4•I-), as shown in Fig. S8."

(10, AR): As alluded to by the referee, ions containing Cl exhibit recognizable patters (most notably at m/z + 2 positions), which was used to verify isolated organic chlorides and chloronitrates in this study. Well-isolated ions consistent with organic chloride, confirmed by the isotopic signals, were identified in the mass spectra. The difficulty with confirming (or denying) larger chloronitrates based on the isotopic signal is that non-chlorinated compounds dominate both the m/z and m/z + 2 positions. In this example, a C$_{10}$ and a C$_{11}$ organic nitrate compound could be fitted in place of the C$_{12}$ chloronitrate. We now more strongly emphasize the uncertainties with organic chloride identification in the revised version of the manuscript, which was the intended purpose of the quoted text.

*(11, RC): P. 6, l. 21 – 22: Did you actually observe that as well if you compare your dry and humid experiments, no? You mention this as a finding in your conclusions as well, but I cannot find it as a result in the manuscript.*

P.6, L.21-22: "Organochloride formation is expected to be lower under humid conditions, where DHF formation is inhibited (Holt et al., 2005; Zhang et al., 2014; Ziemann, 2011).

(11, AR): Based on data presented in the ACPD manuscript, evidence of organochloride suppression was observed for dodecane SOA (and to some extent the decane SOA), where the particulate chloride concentration (measured as HCl$^+$ ion in the ACSM) decreased from Exp. 11 (RH < 5 %) to Exp. 12 (RH ~67 %), as shown in Table 1. Using the FIGAERO-CIMS, we observed similar evidence for dodecane SOA in the terms of the mass fraction of -Cl functional group and the overall mass fraction of organic chlorides and chloronitrates. For octane and decane SOA, no clear evidence was observed. We have clarified this in the main text and added the results obtained using FIGAERO-CIMS.

**Addition to Results and Discussion:** "[…]. Evidence consistent with organochloride suppression under humid condition was observed for dodecane SOA, where the mass fraction of organic chloride decreased from 0.15 (Exp. 11, < 5 % RH) to 0.13 (Exp. 12, 67 % RH) as measured by the FIGAERO-CIMS. The mass fraction of the -Cl functional group decreased from 1.8 E$^{-2}$ to 1.6 E$^{-2}$ as measured by the FIGAERO-CIMS or from 1.4 E$^{-2}$ to 1.1 E$^{-2}$ as measured by the ACSM (see Table 1) as the RH increased. No clear differences were observed for octane or decane SOA, which may be due to the lower organochloride formation, which is especially challenging for chloride quantification using the ACSM, or due to the less extreme RH conditions investigated."

*(12, RC): P. 7, l. 3-4: Are those fragmentation reactions in the particle phase, with subsequent evaporation of the resulting compounds? This would be consistent with the observation of loss of SOA mass (mentioned e.g. on p. 5, l. 21 – 22). Please elaborate. Experiments shown here are all under dry conditions. What about humid conditions?*

P.7, L.3-4: "As oxidation continued, the importance of fragmentation reactions increased relative to that of functionalization reactions (Lambe et al., 2012)."

P. 5, L. 21-22: "Oxidation continued under UV driven by chlorine radicals, and the SOA concentration began to decay due to oxidative fragmentation (Kroll et al., 2011; Lambe et al., 2012; Wang and Hildebrandt Ruiz, 2017)."

(12, AR): We would expect the oxidation and fragmentation to occur primarily in the gas-phase, resulting in the change of particle-phase composition via equilibrium partitioning. Previous study with OH-initiated pentadecane (C$_{15}$) shows that gas-phase chemistry drives oxidation (aided by the dehydration of cyclic hemiacetal, which does occur in the condensed phase), while heterogeneous reactions (e.g. hemiacetal formation) would decrease the aerosol volatility. Heterogenous oxidation of organic and inorganic aerosol has been reported to occur at non-negligible rates (Bertram et al., 2001; George and Abbatt, 2010) but is beyond the scope of this work. SOA decay was also observed under humid conditions. We note that vapor wall loss, which can worsen under more humid conditions (Huang et al., 2018b), would also contribute to the observed SOA concentration decrease in addition to oxidative fragmentation. This has been clarified in the manuscript.

**Revision/Addition to Results and Discussion:** "As oxidation continued, driven primarily by gas-phase chemistry (Aimanant and Ziemann, 2013), the importance of fragmentation reactions increased relative to that of functionalization reactions (Lambe et al., 2012). The heterogenous oxidation of SOA (Bertram et al., 2001; George and Abbatt, 2010), which is expected to drive the oxidation of very large (n>30) alkanes (Lim and Ziemann, 2009b), may also contribute to the oxidation and fragmentation but its impacts are beyond the scope of this work."

*(13, RC): P. 7, l. 13 – 14: It becomes clear after discussion of Figure 5, but it would be helpful for the reader to mention here why you use the temperature range of 40–140 oC*

P.7, L.13-14: "The spectra are calculated from the average desorption ion signals observed when the filter temperature was between 40 and 140 oC."

(13, AR): This has been clarified in the revised manuscript.

**Addition to Results and Discussion:** "[…] As shown below and in Fig. 5, SOA components desorbed mostly effectively in this temperature range, with most organic ions having $T_{max}$ within this temperature range. At desorption temperature above 140 $^{o}$C, inorganic ions began to dominate the spectra."

*(14, RC): P. 8, l. 8 – 10: What do you base your interpretation of "low-temperature thermal fragmentation products" on? I agree that there must be fragmentation, but I am not sure you have enough evidence for that being thermal fragmentation during desorption.*

P.8 L.8-10: "These small organic compounds (detected as I- adducts) were likely low-temperature thermal fragmentation products. Prominence of ions smaller than m/z 127 (i.e. ions generated not from iodide-adduct formation but possibly acid exchange or charge transfer), including Cl- and a range of organic ions, was consistent with low-temperature thermal fragmentation."

(14, AR): Perhaps the term "low-temperature decomposition products" is more appropriate than "low-temperature fragmentation products" in this context. We recognize that some compounds in region 1 (Fig. 5) may correspond to semi-volatile molecular compounds. At the same time, there are compounds detected that would be too volatile to be present in the particle-phase and should therefore be some form of instrument artifact. Because their intensity varied with the desorption temperature (hence the apparent $T_{max}$), they should be thermally induced, which led to the designation of "low-temperature decomposition products." We have clarified this description.

**Revision to Results and Discussion:** "Region 1 (m/z < 350, 40 < Tmax < 90 $^{o}$C) is composed of a group of semi-volatile compound with similar $T_{max}$ values. The region also includes iodide-adduct which correspond to organic species that would be too volatile to be present as molecular compounds in the particle-phase and are likely low-temperature decomposition products. Note that region 1 also include ions not containing I$^{-}$, which could be formed from acid exchange, charge transfer, or potentially ion-induced fragmentation."

*(15, RC): P.9, section 3.3: How reproducible are your thermograms and corresponding Tmax for one compound and stable conditions? This information should be added e.g. to the supplementary section.*

P.9, section 3.3: (summary) Tmax was observed to increase with ammonium sulfate loading for ammonium sulfate decomposition ions. Similar loading dependence was observed for some SOA components for filters collected at different points during an experiment over various durations.

(15, AR): The thermogram reproducibility is validated using aerosols (injected into the chamber) generated from a nebulized 1.2 E$^{-2}$ M levoglucosan aqueous solution. The results are shown Fig. S12. The thermogram shape is fairly reproducible as shown in Fig. S12(a). Similar to

observations made for ammonium sulfate particles, the $T_{max}$ for levoglucosan aerosol also increases with filter loading. A sigmoidal fitting curve is shown to guide the eyes. $T_{max}$ appears to level off towards the higher filter loading range, possibly due to saturation effects (Huang et al., 2018a).

**Addition of Fig. S12**

[Figure]

Fig. S12: (a) Thermogram observed for levoglucosan aerosol at different filter loading conditions, and (b) the correlation between $T_{max}$ and filter loading, where a sigmoidal curve was fitted to guide the eyes. Levoglucosan was generated from a1.2 $E^{-2}$ M aqueous solution.

*(16, RC): P. 11, l. 22 – 23: This sentence is formulated too strongly based on the observations you present in your results section.*

P.11, L.22-23: "Using the ACSM and the FIGAERO-CIMS, trace amounts of alkane-derived organochlorides were observed, produced via chlorine-addition to the heterogeneously produced dihydrofuran compounds."

(16, AR): We have revised the conclusion to reflect the uncertainties with organochloride identification.

**Revision to Conclusion:** "Trace amounts of organochloride were observed using the ACSM and the FIGAERO-CIMS, likely produced via chlorine-addition to heterogeneously produced dihydrofuran compounds. Although the identification of ions consistent with organochloride directly derived from the $C_8$-$C_{12}$ dihydrofurans remain uncertain due to overlapping peaks, small organochloride components could be clearly identified in the HR spectra."

*(17, RC): Technical corrections: P. 6, l. 8: Should be ACSM P. 8, l. 3: Propose*

(17, AR): Technical errors have been corrected.

[revised manuscript text omitted]

---

## Author Response (AR1)

Author Response to Comments by Anonymous Referees

We thank the referees for their suggestions and recommendations. Below are our responses to all comments.

(RC): Referee comment
(AR): Author response

Anonymous Referee #1

*This paper reports results of a laboratory study of secondary organic aerosol (SOA) formation from the oxidation of a series of n-alkanes with Cl atoms in the presence of NOx in a smog chamber. The gas-phase composition of products and SOA was analyzed using a FIGAERO-CIMS to obtain information on product molecular formulas and thermal desorption profiles, and an ACMS and SEMS were used as well. The results were then interpreted in terms of the reasonably well-established mechanism for alkane oxidation by OH radicals, with the addition of possible oligomer-forming reactions. Interesting observations are reported on chlorine-containing products, organic nitrates, other multi-generation oxygenated products, and a new 2-D format is used to present composition and thermal data. The results provide a nice comparison to OH oxidation and the molecular and thermal characterization of products may be useful for future development and testing of models for these reactions. The experiments are very well done, and the introduction, data analysis, and interpretation are thorough and clearly written. Cl oxidation of VOCs has not been well studied, and as shown here its products can differ in significant ways from OH oxidation. There is thus plenty more to be done, but this paper is a good starting point for detailed SOA chemistry studies. I think the paper can be published in ACP after the following comments are addressed.*

We thank the reviewer for the detailed assessment of the manuscript. As suggested and detailed below, we have expanded the discussions on both the gas-phase oxidation chemistry and particle-phase composition.

*(1, RC) In a study of the aging of pentadecane by OH/NOx that was not cited here (Aimanant and Ziemann, AS&T, 2013), evidence was presented based on thermal desorption profiles for the presence of acetal dimers formed from cyclic hemiacetals. These would likely be very stable and so probably desorb intact. Do the authors see any evidence for such oligomers?*

(1, AR): Ions consistent with acetal and hemiacetal dimers as proposed by Aimanant and Ziemann (2013) were identified in the SOA using the FIGAERO-CIMS. In addition, ions consistent with hemiacetal dimers formed via reaction between hydroxycarbonyl nitrate and hydroxy dihydrofuran as proposed by Schilling Fahnestock et al. (2015) were identified. A diagram depicting the potential formation mechanisms, Fig. S11, is added to the S.I. A discussion on oligomerization pathways and identification of oligomeric compounds is added to the main text. Bimodal thermal desorption behaviors observed for numerous ions indicate that some oligomers underwent thermal decomposition. It is difficult to estimate the fraction of oligomer that desorbed intact, given how the decomposition mode (second desorption signal peak in a thermogram) for many ions share the same $T_{max}$, as illustrated in Figs. 5 and S8.

**Addition of S.I. figure:**

**Figure S11:** Oligomerization via the reaction between (a) carbonyl hydroxyl nitrate and hydroxy dihydrofuran proposed by Schilling Fahnestock et al. (2015) and (b) cyclic hemiacetal with either a 1,4-hydroxycarbonyl or another cyclic hemiacetal proposed by Aimanant and Ziemann (2013). Condensed-phase isomerization between acetal and hemiacetal dimers is also possible, as shown in (b).

**Addition to Results and Discussion:**

P.13, L.2 – P.14, L.7: "Oligomerization may proceed via condensed-phase reactions between cyclic hemiacetal compounds, forming acetal dimers, or reactions between cyclic hemiacetal and 1,4-hydroxycaronbyl compounds, forming hemiacetal dimers (Aimanant and Ziemann, 2013). Condensed-phase, acid-catalyzed isomerization of hemiacetal and acetal dimers is also possible (Aimanant and Ziemann, 2013). For OH-initiated oxidation of dodecane under high NO$_x$ conditions, hemiacetals and nitrate hemiacetals are reported to dominate the oligomer composition (Schilling Fahnestock et al., 2015), where nitrate hemiacetals could form via the reactive uptake of hydroxy dihydrofuran and hydroxycarbonyl nitrate compounds (Schilling Fahnestock et al., 2015). The three oligomerization pathways are illustrated in Fig. S11. Ions consistent with acetal, hemiacetal, and nitrate hemiacetal compounds were observed in the particle phase using the FIGAERO-CIMS in regions 2 and 3. Oligomer formation via aldol condensation by highly oxidized compounds is also possible (Schilling Fahnestock et al., 2015). Oligomerization may also proceed through a NO$_3$-elimination, reverse esterification process involving two alkyl nitrate compounds to form a singly nitrate oligomer, which was recently proposed for α-pinene SOA, though the exact mechanism remains elusive (Faxon et al., 2018)."

*(2, RC): Can the composition data be used to estimate average molecular weights for the SOA analyzed by the FIGAERO-CIMS? It would be useful to use these values to convert the mass yields of SOA to molar yields of SOA, since this gives a better indication of the efficiency of SOA formation. Otherwise the large mass yields (due in large part to the extensive oxidation) give the impression that a large fraction of the alkane is converted to SOA, which is not really true.*

(2, AR): Based on FIGAERO-CIMS data, assuming equal sensitivity for all ions, the average molecular weight, mw$_{avg}$ is estimated based on the desorption intensity I$_i$ and the molecular weight, mw$_i$ of all ions identified,

$$mw_{avg}(T_d) = \frac{\sum_i mw_i \times I_i(T_d)}{\sum_i I_i(T_d)}$$

where $mw_{avg}$ and $I_i$ vary with desorption temperature, $T_d$ during a single FIGAERO desorption. A single average molecular weight for the entire FIGAERO desorption run can be calculated based on the integrated values over the $T_d$ range,

$$mw_{SOA} = \frac{\int(\sum_i mw_i \times I_i(T_d))\,\Delta T_d}{\int(\sum_i I_i(T_d))\Delta T_d}$$

where $\Delta T_d$ is the step change in desorption temperature. It should be noted that because of thermal decomposition, which has been observed for monomers as well (Stark et al., 2017), the $mw_{avg}$ and $mw_{SOA}$ calculated using the FIGAERO-CIMS data likely underestimate the actual average SOA molecular weight. For octane SOA, $mw_{SOA}$ calculated in this way ranges from 206 to 226 g mol$^{-1}$; for decane SOA, $mw_{SOA}$ ranges from 241 to 265 g mol$^{-1}$; for dodecane SOA, $mw_{SOA}$ ranges from 260 to 270 g mol$^{-1}$. The molar SOA yield can then be calculated as,

$$Y_{molar} = \frac{n_{SOA}}{\Delta n_{VOC}} = \frac{m_{SOA}/mw_{SOA}}{\Delta m_{VOC}/mw_{VOC}} = Y\frac{mw_{VOC}}{mw_{SOA}}$$

where $n_{SOA}$ and $n_{VOC}$ are the molar concentrations of SOA and VOC. The molecular weight of VOC, $mw_{VOC}$ is known for the alkane precursors. The corresponding molar yield ranges from 0.08 to 0.14 for octane SOA, 0.26 to 0.47 for decane SOA, and 0.72 to 1.04 for dodecane SOA. The above-unity molar SOA yield observed for dodecane (Exp. 9) indicates that SOA mass may be overestimated, which may be the result of uncertainties with the collection efficiency (CE) and relative ionization efficiency (RIE) assumed for the SOA, which are addressed by (7. AR) below in response to (7. RC). The method described above is added to the section S3 in the S.I. Results are tabulated in Table S1.

*(3, RC): Is there any indication for chemistry occurring between Cl, O2, and NOx? Would this be expected?*

(3, AR): Ions consistent with Cl and/or $NO_x$ oxidation products, including $HNO_3$, ClONO (or $ClNO_2$), ClO, HONO, $HO_2NO_2$, and $N_2O_5$ were observed with the I$^-$ CIMS, which are consistent with literature findings. Selected time-series are added to Fig. 1 in the main text and to Fig. S12 in the S.I.. The results suggest that secondary $HO_x$ chemistry and perhaps $NO_3^-$ chemistry (in the dark) may be important, which warrant future investigations. Discussion of the interactions between Cl, $O_3$, and $NO_x$ are added to the main text.

**Updates to Fig. 1:**

[Figure]

**Fig 1.** Representative trends of SOA and trace gas species during the photooxidation period. Data from dodecane oxidation (Exp. 11) are shown. Particulate chlorine concentrations were multiplied by 70 for ease of comparison. Observation of select ions such as HONO and $HO_2NO_2$ are indicative of secondary $HO_x$ chemistry. Formation of ClONO from the interaction between Cl$^{\bullet}$ and $NO_2$ was also observed.

**Addition to the S.I.:**

Formation of $HO_2NO_2$ likely proceeds via

$$HO_2 + NO_2 \xrightarrow{M} HO_2NO_2 \qquad\qquad \text{R. (S1)}$$

Formation of HONO under UV can proceed via

$$HO_2 + NO \rightarrow OH + NO_2 \qquad\qquad \text{R. (S2)}$$
$$OH + NO \rightarrow HONO \qquad\qquad \text{R. (S3)}$$

The trends for $HO_2NO_2$, HONO, and select gas-phase species observed by the I$^-$ CIMS are shown in Fig. S12.

[Figure]

**Fig. S12:** Representative trends of SOA and trace gas species during the photooxidation period. Data from dodecane oxidation (Exp. 11) are shown, similar to Fig. 1 in the main text. Additional species shown include $ClONO_2$ (from reaction between ClONO with $NO_2$, Lesar et al., 2006), ClO (possibly a ion fragment of ClONO), HCl (from H-abstraction by Cl$^{\bullet}$ or perhaps Cl-elimination reactions), and $HNO_3$ (due to the oxidation of $NO_2$ by OH).

**Addition to Results and Discussion:**
P.8, L.12 – 19: "In addition, ions consistent with $HO_2NO_2$, HONO, and $ClNO_2$ were observed in the gas phase, as shown in Fig. 1. The $ClNO_2I^-$ ions are expected to correspond to both $ClNO_2$ (< 20 %) and chlorine nitrite (ClONO, > 80 %) from the reaction between $Cl^•$ and $NO_2$ (Golden, 2007; Niki et al., 1978). Formation of $HO_2NO_2$ from the reaction of $HO_2$ and $NO_2$ has been observed using $I^-$ CIMS previously (Veres et al., 2015). Formation of HONO was likely due to the interactions between $HO_x$ and NO (see R. S2 and R. S3 in the SI). Under UV, HONO decays to background level. Production of $HO_2NO_2$ and HONO is indicative of secondary $HO_x$ chemistry enabled by primary Cl-alkane oxidation chemistry, consistent with previous studies (Wang and Hildebrandt Ruiz, 2017; Young et al., 2014)."

*(4, RC): Structure-activity relationships for the reactions of alkanes with Cl atoms indicate that the ratio of rate constants for abstraction of primary/secondary H-atoms is significantly smaller than for OH reactions, indicating that more reaction occurs at the ends of the alkanes with Cl than with OH. Have the authors considered what the consequences of this might be with regards to the products being formed?*

(4, AR): With increased terminal H-abstraction by Cl radicals as compared to OH-radicals, we would expect increased primary alkyl nitrate formation. This should result in lowered saturation vapor pressure of the oxidation products (Lim and Ziemann, 2009b; Yeh and Ziemann, 2014, 2015), consistent with the higher SOA yield observed here. We have added a discussion of this factor to the main text.

**Addition to Introduction:**
P.3, L.9 -12: "According to structure-activity relationships, terminal hydrogen abstraction occurs more frequently with alkane-Cl than with alkane-OH reactions (Kwok and Atkinson, 1995), resulting in different product distributions; for example, increased formation of primary alkyl nitrates."

**Addition to Results and Discussion**:
P.9, L.14 – 17: "Formation of primary alkyl nitrates due to terminal H-abstraction by Cl could contribute to the higher SOA yields observed for alkane-Cl than for alkane-OH oxidation, as primary alkyl nitrates have lower saturation vapor pressures than secondary alkyl nitrates (Lim and Ziemann, 2009b; Yeh and Ziemann, 2014, 2015)."

*(5, RC): Have the authors considered comparing vapor pressures estimated from the measured desorption temperatures with those predicted from proposed molecular structures and the SIMPOL group contribution method?*

(5, AR): Previous FIGAERO-CIMS studies show that vapor pressures calculated using either the $C^*$-$T_{max}$ correlation or the direct partitioning method (Lopez-Hilfiker et al., 2015) can have significant disagreements with SIMPOL results (D'Ambro et al., 2017; Stark et al., 2017), which have been mostly attributed to the effects of thermal decomposition. Another complication with $C^*$ estimation can also arise from potential filter loading dependence for $T_{max}$ (Huang et al., 2018

and this work). In our opinion, this requires further investigation before vapor pressure measurements by FIGAERO and SIMPOL may be reconciled.

*(6, RC): It would be useful to provide a little more detailed speculation on the proposed nature of the thermal decomposition products. Are these thought to be formed by simple reversible dissociation of hemiacetals, for instance, or by cleavage of strong C-C or C-O bonds? Or might they just be compounds that are "trapped" in a solid SOA matrix that cannot escape until the SOA melts?*

(6, AR): Given that numerous organic ions exhibit bimodal thermal desorption behavior, it is reasonable to attribute it to the reversible dissociation of oligomers, a phenomenon that has been observed in the FIGAERO at room temperatures (Schobesberger et al., 2018) and at high desorption temperatures (Faxon et al., 2018). Decomposition of pure monomeric organic acids has also been reported previously, indicating that cleavage of C-C and C-O bonds (C-OH or C-OOH) does occur in the FIGAERO (Stark et al., 2017). Trapping of molecular compounds inside the SOA matrix may be possible if the aerosol existed in a "glassy state", and could result in a sudden release of volatilized vapors (all having similar $T_{max}$, as seen in region 1 and 3) though the aerosol phase state cannot be evaluated based on data available. It is worth noting that ions associated with ammonium sulfate decomposition all share the same $T_{max}$ with or without SOA present (that is, uniform $T_{max}$ can be the result of thermal decomposition as well).

**Addition to Results and Discussion**:
P.13, L.28 – P.14, L.4: "The thermal decomposition products may be the result of reversible dissociation of oligomers (Faxon et al., 2018; Schobesberger et al., 2018) or the cleavage of C-O or C-C bonds, which has been observed in the thermal decomposition of monomeric organic acids (Stark et al., 2017). Additionally, SOA may exist in a "glassy" state (Koop et al., 2011), trapping volatile components (Perraud et al., 2012) that are released as the SOA mass is depleted."

*(7, RC): I did not see a quoted density for calculating SOA yields from SEMS data. This value and its source should be noted.*

(7, AR): The SOA yield was calculated from the ACSM measurement of organic aerosol concentration, assuming a collection efficiency, CE of 0.5 and a relative ionization efficiency, RIE of 1.4. The average, apparent SOA density calculated using ACSM and SEMS data is around $2.1 \pm 0.3$ g cm$^{-3}$, which is higher than the densities previously reported for OH-alkane SOA formed under high NO$_x$ conditions, 1.06 to 1.28 g cm$^{-3}$ (Lim and Ziemann, 2009a; Loza et al., 2014). The discrepancy can result from the differences in SOA composition, aerosol oxidation state (Kuwata et al., 2012), or from instrument uncertainties. For instance, studies report that RIE of organics may depend on the aerosol oxidation state of carbon (Li et al., 2018; Xu et al., 2018). For $-1 < OS_C < 0.5$, which overlaps with the average $OS_C$ as indicated by the FIGAERO-CISM measurement as shown in in Fig. 4d, the average RIE is reportedly $1.6 \pm 0.5$ instead of 1.4 (Xu et al., 2018), which is higher than the RIE value of 1.4 assumed here. Re-calculation using the updated RIE value would lower the apparent SOA density closer to agreement with previous studies. Without an independent measurement for particle mass (e.g.

using a particle mass analyzer), it is not possible to ascertain the exact SOA density. Additional discussion associated with SOA quantification are added to the main text.

**Additions to Methods:**
P.5, L.23 – 25: "Organic aerosol concentrations were calculated using ACSM measurements assuming a collection efficiency of 0.5 and RIE of 1.4, corrected for depositional particle wall loss…[]"

**Additional to Results of Discussion:**
P.9, L.17 – 26: "The apparent SOA density, calculated using the ACSM SOA mass measurement and the SEMS SOA volume measurement, was $2.1 \pm 0.3$ g cm$^{-3}$, which is substantially higher than that reported for OH-alkane SOA formed under high NO$_x$ conditions, 1.06 to 1.28 g cm$^{-3}$ (Lim and Ziemann, 2009a; Loza et al., 2014). These differences could be due to uncertainties associated with the ACSM and similar instruments, specifically the RIE (Li et al., 2018; Xu et al., 2018) and CE (Docherty et al., 2013; Middlebrook et al., 2012; Robinson et al., 2017), which may vary with SOA composition, oxidation state, or phase state. Assuming an SOA density of 1.06 g cm$^{-3}$, the SOA yield (0.10 to 0.99) is still higher for Cl-alkane oxidation as compared to OH-alkane oxidation. Molar SOA yields, calculated using the average molecular weight of species identified with the FIGAERO-CIMS, are summarized in Table S1."

**Anonymous Referee #2**

(AR): Lines specified by the referee is copied below from the ACPD manuscript; revisions are noted in the author response

*(1, RC) Wang et al. present results from environmental chamber experiments of C8-C12 alkane oxidation initiated by chlorine under dry/humid and high NOx conditions. They show that yields are higher than for OH-initiated oxidation of the respective alkanes, indicative of the importance of these reactions for SOA production. Identified compounds include organonitrates, organochlorides. FIGAERO-CIMS data are used to look into the connection of chemistry and volatility via a new way of representing thermograms. Overall, this is a very well written paper, and the study and interpretation of results are sound. I therefore recommend this paper to be published after minor revisions. Apart from a few specific comments (see below), I have two general comments regarding the manuscript.*

*First, the manuscript would profit from a few more lines on its atmospheric relevance. Using alkanes as VOC precursors, and performing experiments under high NOx conditions indicates the authors wanted to simulate an urban/polluted atmosphere. A short discussion on this, including the importance of Cl- oxidation in such environments, as well as the choice of RH conditions, would give the study more (atmospheric) importance.*

(1, AR): We thank the referee for the recommendations. We have added some discussions on the experimental design choices and the atmospheric relevance of Cl oxidation chemistry

**Revisions / Additions to Introduction**:
P.2, L.26 – P.3, L.7: "$ClNO_2$ photolysis in the early morning produces Cl and $NO_x$, which has been shown to enhance $RO_2$ production from alkane oxidation in near coastal regions (Riedel et al., 2012) as well as OH radical propagation in urban environments (Young et al., 2014). In addition to reactive chlorine emissions from water treatment (Chang et al., 2001) and fuel combustion (Osthoff et al., 2008; Parrish et al., 2009), the rising usage of volatile chemical products (VCP) such as pesticides, cleaning products, and personal care products may be a significant source of reactive chlorine compounds and VOCs in urban environments (Khare and Gentner, 2018; McDonald et al., 2018). VOC-Cl oxidation products such as isomers of 1-chloro-3-methyl-3-butene-2- (CMBO), a tracer for isoprene-Cl chemistry (Nordmeyer et al., 1997), have been observed in highly polluted environments (Le Breton et al., 2018; Tanaka et al., 2003). "

**Addition to Methods**:
P.4, L.15 – 18: Under typical atmospheric conditions, elevated RH can be expected, especially within the marine boundary layer in near-coastal regions, where Cl-alkane chemistry may be important (Riedel et al., 2012). Therefore, SOA formation under humid conditions was also investigated.

**Addition to Conclusion**:
P.19, L.15 – 17: Overall, these results show that chlorine-alkane oxidation could be an important pathway for SOA production and ageing, especially in highly polluted environments replete with alkane, $NO_x$, and reactive chlorine emission sources.

*(2, RC): Second, there is somewhat a disconnect in the narrative between section 3.1 and section 3.2, which also represents a disconnect between ACSM and FIGAERO-CIMS data. I suggest the authors try to connect these two parts better. Section 3.1 (SOA and organic chloride formation) is entirely ACSM based. Why? Why were FIGAEROCIMS data not used? Some statements made in a tentative manner could be confirmed/ looked into using FIGAERO-CIMS data (see specific comments below).*

(2, AR): The revised manuscript makes more use of quantitative FIGAERO-CIMS data, and compares ACSM to FIGAERO data. Additional gas- and particle-phase results are presented in the response to the following questions.

*(3, RC): P. 1, l. 26/ p. 2 l. 11: Where is chlorine oxidation important? See comment above, the paper would benefit from a few more lines on its atmospheric relevance.*

P.1 L.26: "Evaporation of POA due to dilution can provide additional gas-phase alkanes, which can undergo photooxidation initiated by OH, NO3, as well as chlorine radicals (Aschmann and Atkinson, 1995; Atkinson and Arey, 2003)."

P.2 L.11: "Recent field studies have identified reactive chlorine compounds in diverse locales from natural and anthropogenic sources (Faxon and Allen, 2013; Finlayson-Pitts, 2010; Saiz-Lopez and von Glasow, 2012; Simpson et al., 2015)."

(3, AR): We have expanded the discussion on the conditions under which chlorine chemistry may be important as suggested as detailed in (1, AR).

*(4, RC): P. 2, l. 32 –33: You specifically mention here low NOx, as presumably the cited study was done under such. What are then the implications for your study? I suggest reformulating this sentence.*

P.2. L.32-33: "Chlorine radicals can react with dihydrofuran via both H-abstraction and Cl-addition, producing chlorinated (e.g. dichlorotetrahydrofurans) and non-chlorinated compounds (e.g. furanones) under low NOx conditions (Alwe et al., 2013)."

(4, AR): The referenced study was performed under low NOx conditions. The main implication here is that the reaction between Cl and dihydrofuran can produce both chlorinated and non-chlorinated products. Under high NOx conditions, both chlorinated and non-chlorinated organonitrate compounds may be formed. The following sentence has been added to clarify the implication of the NOx study and its connection with the high NOx scenario.

**Addition to Introduction:**
P.4, L.4 - 6: "…under low NOx condition (Alwe et al. 2013). Similarly, formation of both chloronitrates (via Cl-addition) and organonitrates (via H-abstraction) from alkane-Chl oxidation is possible in the presence of NOx."

*(5, RC): P. 4, l. 28: UMR, why not HR? In Figure 5 you present molecular formulae of compounds, indicating HR analysis. Please clarify.*

P.4, L.28: "The 2-D thermogram is comprised of normalized unit-mass resolution 1-D thermograms, each expressed as a percentage color scale of the maximum desorption signal."

(5, AR): We have conducted HR analysis and have experimented with a HR 2-D thermogram as the referee alluded to. We decided to present the UMR version due to its simplicity, where the multimodal thermal desorption behavior and a general $T_{max}$-molecular weight relationship can be visualized in the absence of explicit molecular formulae, which is especially helpful for studying the desorption behavior of high molecular weight compounds which may have a vast number of possible molecular makeups. Another practical concern with the HR 2-D thermogram is the uneven spacing of the molecular weight of the identified ions, which presents some technical issues for plotting on a heat map. We have added a brief discussion on the advantages and disadvantages of using UMR vs HR data for constructing the 2-D thermogram.

**Change / Addition to Methods**:
P.5, L.1 – 9: "The 2-D thermogram is comprised of normalized unit-mass resolution 1-D thermograms, each expressed as a percentage color scale of the maximum desorption signal. 2-D thermogram applications are discussed in Section 3.2. The advantage of using UMR over HR data is the ability to investigate the SOA thermal desorption behavior over the entire *m/z* and volatility (i.e. $T_{max}$) range without having to assign chemical formulae to all ion. This HR analysis is time-consuming especially for high molecular weight compounds whose exact molecular composition can be difficult to ascertain. The disadvantage of using UMR over HR data is the overlapping of ions and potential interference by isotopic signals or non-adduct ions. Therefore, the 2-D thermogram should be used as complement rather than a replacement of the HR analysis."

*(6, RC): P. 5, l. 11: This could be confirmed with FIGAERO-CIMS data. Why were they not added?*

P.5, L.11: "Given the same oxidation conditions, SOA products derived from longer alkane precursors appeared less oxidized…"

(6, AR): We have added Fig. 4(d) to support this observation using the FIGAERO-CIMS data, which shows that the SOA oxidation state decreases with increasing precursor length in agreement with the ACSM data analysis as shown in Table 1 and Fig. S1. Figure 4(b) in the ACPD manuscript is now Fig. 4(c).

**Addition to Methods:**

P. 7, L.8 – 16: "Based on the SOA molecular composition as observed by the FIGAERO-CIMS, the average oxidation state of carbon (OS$_C$) may be estimated,

$$OS_C = 2 \times O:C - H:C + NO_3:C + Cl:C \qquad \text{Eq. (1)}$$

where NO$_3$:C, Cl:C, O:C, and H:C are the molecular ratios of the number of -NO$_3$ functional groups, -Cl functional groups, non-NO$_3$ oxygen atoms, and H atoms to the number of carbon atoms for any given compound. The average SOA OS$_C$ is calculated based on iodide-adducts only. For simplicity, all organic ions were assumed to have equal sensitivity, which is known to vary with ion cluster binding energy and sample RH (Hyttinen et al., 2018; Iyer et al., 2016; Lopez-Hilfiker et al., 2016)."

**Addition of Fig. 4(d):**

[Figure]

**Figure 4d.** Average oxidation state of carbon (OS$_C$) and carbon number of ions observed during the temperature-programmed desorption of SOA produced from the Cl-initiated oxidation of octane (Exp. 3), decane (Exp. 7), and dodecane (Exp. 11). Under similar oxidant conditions, SOA derived from longer VOC precursor were on average less oxidized.

*(7, RC): P. 5, l. 18 – 19: I⁻ should cluster with Cl$_2$. Why do you use the Cl⁻ signal to track Cl$_2$?*

P.5, L.18-19: The Cl$_2$ concentration was estimated using I⁻ CIMS by tracking the Cl⁻ ion.

(7, AR): Cl⁻ ion was used at one point during the drafting of the manuscript. The Cl$_2$I⁻ ion was ultimately used in Fig. 1 shown in the discussion paper but the text was not updated by mistake. This has been corrected.

P.8, L.11 – 12: "The Cl$_2$ concentration was estimated using I⁻ CIMS by tracking the Cl$_2$I⁻ ion"

*(8, RC): Please clarify. P. 5, l. 21 – 22: Did you see any evidence of that in FIGAERO-CIMS gas-phase measurements?*

P.5, L.21-22: "Oxidation continued under UV driven by chlorine radicals, and the SOA concentration began to decay due to oxidative fragmentation (Kroll et al., 2011; Lambe et al., 2012; Wang and Hildebrandt Ruiz, 2017)."

(8, AR): We have added Fig. 4(b) to show the simultaneous increases in the average $OS_C$ and decreases in the $n_C$ of gas-phase species as observed the $I^-$ CIMS over the course of the photooxidation (0 to 60 mins), which are consistent with oxidative fragmentation. Results also show that octane oxidation products are significantly more oxidized compared to decane and dodecane oxidation products.

**Addition of Fig. 4(b):**

[Figure]

Figure 4(b). Average $OS_C$ and $n_C$ of gas-phase compounds observed during the photooxidation period for Exp. 3 (Octane), 7 (decane), and 11 (dodecane). As oxidation continues, $OS_C$ increased and $n_C$ decreased, consistent with oxidative fragmentation.

*(9, RC): P. 6, l. 1-2: Can you confirm that with the organonitrate measurements of the FIGAEROCIMS?*

P.6, L.1-2: "Higher initial NO concentrations led to higher SOA yields and lower ozone production for all precursors, as shown in Table 1 and Fig. S2. This is similar to alkane OH SOA formation, where higher NO concentrations lead to more abundant organic nitrate formation, which increases the SOA volume (Schilling Fahnestock et al., 2015) and density (Loza et al., 2014)."

(9. AR): FIGAERO data do show increased organonitrate abundance in the particle phase for SOA formed under NO-only conditions as compared to $NO_2$-only conditions in terms of both molar ratio (i.e. average number of $-NO_3$ functional group per molecule) and mass fraction (i.e. mass of organonitrate as a fraction of the identified SOA mass). For mixed NO-$NO_2$ conditions, the observed organonitrate molar ratio and mass fraction are closer to that observed under the $NO_2$ only condition. The results are incorporated into the main text. The tabulated data are added to the S.I.

**Addition to Results and Discussion:**
P.9, L.5 – 10: "[…]This is consistent with FIGAERO-CIMS results where the particulate organonitrate molar ratio (calculated as the average number of $-NO_3$ per molecule) and mass fraction were the highest in SOA produced under NO-only conditions (see Table S1). The organonitrate molar ratio also increased with the alkane precursor length, from 0.57-0.64 for

octane to 0.75-0.93 for dodecane. A similar trend was observed for the orgnaonitrate mass fraction, which increased from 0.53-0.58 for octane to 0.66-0.72 for dodecane."

*(10, RC): P. 6, l. 16 – 18: Also here the isotope signal should help you. Without that, the chloronitrate peak cannot be identified (based on Figure S8).*

P. 6, L. 16-18: "Compounds resembling chloronitrates (e.g. ONO2 C12H18ClO2•I- for dodecane) were tentatively identified in the particle-phase using the FIGAERO but they were not well separated from the shoulder of nearby organonitrate peaks (e.g. ONO2 C12H21O4•I-), as shown in Fig. S8."

(10, AR): As alluded to by the referee, ions containing Cl exhibit recognizable patters (most notably at m/z + 2 positions), which was used to verify isolated organic chlorides and chloronitrates in this study. Well-isolated ions consistent with organic chloride, confirmed by the isotopic signals, were identified in the mass spectra. The difficulty with confirming (or denying) larger chloronitrates based on the isotopic signal is that non-chlorinated compounds dominate both the m/z and m/z + 2 positions. In this example, a $C_{10}$ and a $C_{11}$ organic nitrate compound could be fitted in place of the $C_{12}$ chloronitrate. We now more strongly emphasize the uncertainties with organic chloride identification in the revised version of the manuscript, which was the intended purpose of the quoted text.

**Change / Revision to S.I.:**

[Figure]

**Figure S9:** High resolution fitting at *m/z* (a) 434 and (b) 436 for FIGAERO-CIMS data from Exp. 11. The $C_{12}$ chloronitrate (ONO$_2$-C$_{12}$H$_{18}$ClO$_3$•I$^-$) peak is tentatively identified in (a) but it overlaps with the nearby stronger organonitrate (ONO$_2$-C$_{12}$H$_{21}$O$_5$•I$^-$) peak. Because the chloronitrate peak is a weaker peak with significant overlap with a stronger peak, quantitative assessment would be challenging due to peak fitting uncertainties (Cubison and Jimenez, 2015). The presence of chloronitrate cannot be confirmed or rejected based on its unique isotopic signature at *m/z*+2 positions, as shown in (b), where nonchlorinated compounds also dominate.

*(11, RC): P. 6, l. 21 – 22: Did you actually observe that as well if you compare your dry and humid experiments, no? You mention this as a finding in your conclusions as well, but I cannot find it as a result in the manuscript.*

P.6, L.21-22: "Organochloride formation is expected to be lower under humid conditions, where DHF formation is inhibited (Holt et al., 2005; Zhang et al., 2014; Ziemann, 2011).

(11, AR): Based on data presented in the ACPD manuscript, evidence of organochloride suppression was observed for dodecane SOA (and to some extent the decane SOA), where the particulate chloride concentration (measured as $HCl^+$ ion in the ACSM) decreased from Exp. 11 (RH < 5 %) to Exp. 12 (RH ~67 %), as shown in Table 1. Using the FIGAERO-CIMS, we observed similar evidence for dodecane SOA in the terms of the mass fraction of -Cl functional group and the overall mass fraction of organic chlorides and chloronitrates. For octane and decane SOA, no clear evidence was observed. We have clarified this in the main text and added the results obtained using FIGAERO-CIMS.

**Addition to Results and Discussion:**
P.10, L.18 – 26: "[…].Evidence consistent with organochloride suppression under humid condition was observed for dodecane SOA only, where the organochloride (including chloronitrates) mass fraction decreased from 0.15 (Exp. 11, < 5 % RH) to 0.13 (Exp. 12, 67 % RH) as measured by the FIGAERO-CIMS. The mass fraction of the -Cl functional group decreased from $1.8 E^{-2}$ to $1.6 E^{-2}$ as measured by the FIGAERO-CIMS (Table S1) or from $1.4 E^{-2}$ to $1.1 E^{-2}$ as measured by the ACSM (Table 1) as the RH increased. No clear differences were observed for octane or decane SOA, which may be due to the less extreme RH conditions investigated, uncertainties with organochloride ion identification in the CIMS, or the lower organochloride concentrations observed in Exps.1-8, which is especially challenging for chloride quantification using the ACSM."

*(12, RC): P. 7, l. 3-4: Are those fragmentation reactions in the particle phase, with subsequent evaporation of the resulting compounds? This would be consistent with the observation of loss of SOA mass (mentioned e.g. on p. 5, l. 21 – 22). Please elaborate. Experiments shown here are all under dry conditions. What about humid conditions?*

P.7, L.3-4: "As oxidation continued, the importance of fragmentation reactions increased relative to that of functionalization reactions (Lambe et al., 2012)."

P. 5, L. 21-22: "Oxidation continued under UV driven by chlorine radicals, and the SOA concentration began to decay due to oxidative fragmentation (Kroll et al., 2011; Lambe et al., 2012; Wang and Hildebrandt Ruiz, 2017)."

(12, AR): We would expect the oxidation and fragmentation to occur primarily in the gas-phase, resulting in the change of particle-phase composition via equilibrium partitioning. Previous study with OH-initiated pentadecane ($C_{15}$) shows that gas-phase chemistry drives oxidation (aided by the dehydration of cyclic hemiacetal, which does occur in the condensed phase), while heterogeneous reactions (e.g. hemiacetal formation) would decrease the aerosol volatility. Heterogenous oxidation of organic and inorganic aerosol has been reported to occur at nonnegligible rates (Bertram et al., 2001; George and Abbatt, 2010) but is beyond the scope of this work. SOA decay was also observed under humid conditions. We note that vapor wall loss, which can worsen under more humid conditions (Huang et al., 2018b), would also contribute to the observed SOA concentration decrease in addition to oxidative fragmentation. This has been clarified in the manuscript.

**Revision/Addition to Results and Discussion:**
P.11, L.18 – 23: "As oxidation continued, driven primarily by gas-phase chemistry (Aimanant and Ziemann, 2013), the importance of fragmentation reactions increased relative to that of functionalization reactions (Lambe et al., 2012). The heterogeneous oxidation of SOA (Bertram et al., 2001; George and Abbatt, 2010), which is expected to drive the oxidation of very large (n>30) alkanes (Lim and Ziemann, 2009b), may also contribute to oxidation and fragmentation observed here, but its impacts are beyond the scope of this work."

*(13, RC): P. 7, l. 13 – 14: It becomes clear after discussion of Figure 5, but it would be helpful for the reader to mention here why you use the temperature range of 40–140 oC*

P.7, L.13-14: "The spectra are calculated from the average desorption ion signals observed when the filter temperature was between 40 and 140 oC."

(13, AR): This has been clarified in the revised manuscript.

**Addition to Results and Discussion:**
P.12, L.6 – 9: "[…] As shown below and in Fig. 5, SOA components desorbed mostly effectively in this temperature range, with most organic ions having $T_{max}$ within this temperature range. At desorption temperature above 140 $^{\circ}$C, inorganic ions began to dominate the spectra."

*(14, RC): P. 8, l. 8 – 10: What do you base your interpretation of "low-temperature thermal fragmentation products" on? I agree that there must be fragmentation, but I am not sure you have enough evidence for that being thermal fragmentation during desorption.*

P.8 L.8-10: "These small organic compounds (detected as I- adducts) were likely low-temperature thermal fragmentation products. Prominence of ions smaller than m/z 127 (i.e. ions generated not from iodide-adduct formation but possibly acid exchange or charge transfer), including Cl- and a range of organic ions, was consistent with low-temperature thermal fragmentation."

(14, AR): Perhaps the term "low-temperature decomposition products" is more appropriate than "low-temperature fragmentation products" in this context. We recognize that some compounds in region 1 (Fig. 5) may correspond to semi-volatile molecular compounds. At the same time, there are compounds detected that would be too volatile to be present in the particle-phase and should therefore be some form of instrument artifact. Because their intensity varied with the desorption temperature (hence the apparent $T_{max}$), they should be thermally induced, which led to the designation of "low-temperature decomposition products." We have clarified this description.

**Revision to Results and Discussion:**

P.13, L14 – 21: "Region 1 ($m/z < 350$, $40 < T_{max} < 90\ ^oC$) was composed of a group of semi-volatile compounds with similar $T_{max}$ values. This region also includes iodide-adducts which correspond to species that are too volatile to be present as molecular compounds in the particle phase and are likely low-temperature decomposition products. Prominence of ions smaller than $m/z$ 127 including Cl$^-$ and a range of organic ions, could be acid exchange or charge transfer products (as opposed to I$^-$ adducts) of low-temperature decomposition products."

*(15, RC): P.9, section 3.3: How reproducible are your thermograms and corresponding Tmax for one compound and stable conditions? This information should be added e.g. to the supplementary section.*

P.9, section 3.3: (summary) Tmax was observed to increase with ammonium sulfate loading for ammonium sulfate decomposition ions. Similar loading dependence was observed for some SOA components for filters collected at different points during an experiment over various durations.

(15, AR): The thermogram reproducibility is validated using aerosols (injected into the chamber) generated from a nebulized 1.2 E$^{-2}$ M levoglucosan aqueous solution. The results are shown in the updated Fig. 6. The thermogram shape is fairly reproducible as shown in Fig. 6(a). Similar to observations made for ammonium sulfate particles (now Fig. S7), the $T_{max}$ for levoglucosan aerosol also increases with filter loading. A sigmoidal fitting curve is shown to guide the eyes. $T_{max}$ appears to level off towards the higher filter loading range, possibly due to saturation effects (Huang et al., 2018a).

**Update of Fig. 6**

[Figure]

Fig. 6: (a) The 1-D thermograms for levoglucosan (C$_6$H$_{10}$O$_5$) at different loading conditions and (b) the correlation between filter loading and $T_{max}$. Levoglucosan aerosol was generated by nebulizing a 1.2 E$^{-2}$ M aqueous solution. The aerosol was injected into the clean Teflon chamber, collected onto the FIGAERO filter, and analyzed. The $T_{max}$-loading correlation for pure levoglucosan could be described by a sigmoid function, leveling off at 41 and 65 $^oC$ under very low and very high loading conditions, respectively.

*(16, RC): P. 11, l. 22 – 23: This sentence is formulated too strongly based on the observations you present in your results section.*

P.11, L.22-23: "Using the ACSM and the FIGAERO-CIMS, trace amounts of alkane-derived organochlorides were observed, produced via chlorine-addition to the heterogeneously produced dihydrofuran compounds."

(16, AR): We have revised the conclusion slightly to reflect the uncertainties with organochloride identification.

**Revision to Conclusion:**
P.19, L.9 – 11: "Trace amounts of alkane-derived organochlorides were observed using the ACSM and the FIGAERO-CIMS, likely produced via chlorine-addition to the heterogeneously produced dihydrofuran compounds."

*(17, RC): Technical corrections: P. 6, l. 8: Should be ACSM P. 8, l. 3: Propose*

(17, AR): Technical errors have been corrected.

[revised manuscript text omitted]